# Molecular and cellular evidence for the impact of a hypertrophic cardiomyopathy-associated RAF1 variant on the structure and function of contractile machinery in bioartificial cardiac tissues

Saeideh Nakhaei-Rad[1,2,17], Fereshteh Haghighi[1,3,4,17], Farhad Bazgir [1,17], Julia Dahlmann[4,5], Alexandra Viktoria Busley[4,6,7], Marcel Buchholzer [1], Karolin Kleemann[3,4], Anne Schänzer[8], Andrea Borchardt[9], Andreas Hahn[10], Sebastian Kötter[11], Denny Schanze [5], Ruchika Anand[9], Florian Funk[12], Annette Vera Kronenbitter[12], Jürgen Scheller[1], Roland P. Piekorz[1], Andreas S. Reichert[9], Marianne Volleth [5], Matthew J. Wolf[13], Ion Cristian Cirstea [14], Bruce D. Gelb [15], Marco Tartaglia[16], Joachim P. Schmitt[12], Martina Krüger[11], Ingo Kutschka[3,4], Lukas Cyganek[4,6,7], Martin Zenker [5✉], George Kensah [3,4✉] & Mohammad R. Ahmadian [1✉]

Noonan syndrome (NS), the most common among RASopathies, is caused by germline variants in genes encoding components of the RAS-MAPK pathway. Distinct variants, including the recurrent Ser257Leu substitution in RAF1, are associated with severe hypertrophic cardiomyopathy (HCM). Here, we investigated the elusive mechanistic link between NS-associated RAF1[S257L] and HCM using three-dimensional cardiac bodies and bioartificial cardiac tissues generated from patient-derived induced pluripotent stem cells (iPSCs) harboring the pathogenic *RAF1* c.770 C > T missense change. We characterize the molecular, structural, and functional consequences of aberrant RAF1–associated signaling on the cardiac models. Ultrastructural assessment of the sarcomere revealed a shortening of the I-bands along the Z disc area in both iPSC-derived RAF1[S257L] cardiomyocytes and myocardial tissue biopsies. The aforementioned changes correlated with the isoform shift of titin from a longer (N2BA) to a shorter isoform (N2B) that also affected the active force generation and contractile tensions. The genotype-phenotype correlation was confirmed using cardiomyocyte progeny of an isogenic gene-corrected RAF1[S257L]-iPSC line and was mainly reversed by MEK inhibition. Collectively, our findings uncovered a direct link between a RASopathy gene variant and the abnormal sarcomere structure resulting in a cardiac dysfunction that remarkably recapitulates the human disease.

A full list of author affiliations appears at the end of the paper.

In Noonan syndrome (NS), the most prevalent disease entity among the RASopathies, the development of HCM is associated with significant morbidity and risk of cardiac death. Notably, more than 90% of NS individuals with a *RAF1* variant are affected by HCM, whereas the overall frequency of hypertrophic cardiomyopathy (HCM) in NS is only about 20%[1,2]. One variant, c.770 C > T (p.Ser257Leu), accounts for more than 50% of cases with *RAF1*-associated NS (NSEuroNet database). Similar to other variants affecting transducers and positive modulatory proteins with a role in the MAPK signaling cascade, the Ser257Leu amino acid substitution confers a functional enhancement (gain-of-function (GoF)) to RAF1 kinase activity[3,4]. Functional studies on RAF1 activity have mainly focused on its role in the activation of the MAPK pathway in cancer[5]. Recently, new paths of RAF1 cardiac-specific signaling toward ERK5 and calcineurin–NFAT have been discovered[6–8]. However, how the coaction of RAF1-specific signaling pathways in cardiomyocytes (CMs) results in the phenotypical changes that lead to the development of HCM remains an open question. At the cellular level, HCM is characterized by an increase in CM size, reactivation of the fetal gene program, change in the amplitudes of calcium ($Ca^{2+}$) transients, increased protein synthesis, and changes in the organization of the sarcomere structure and dysfunctional contractility[9]. Therefore, there is a critical need to combine signaling, phenotypical, and functional studies to dissect and understand the pathogenesis of HCM driven by RAF1[S257L] GoF.

To analyze the cardiac-specific function of RAF1 signaling, highly pure populations of human CMs are required to establish informative in vitro HCM disease models, where CMs should be able to respond to external stimuli and stress conditions as well as increase cell size rather than cell number. CMs generated from patient-specific induced pluripotent stem cells (iPSCs) have been used as human HCM models of several genetic etiologies as well as for drug screening[10–14]. Such in vitro models may recapitulate several HCM features at the cellular level, including the increase in cell size, aberrant calcium handling, reactivation of the fetal gene programs, and arrhythmia[7,8,10,15,16].

In this study, we generated and used two different three-dimensional (3D), human cardiac cell models, cardiac bodies (CBs), and bioartificial cardiac tissues (BCTs) in the presence and absence of a MEK inhibitor (PD0325901) to elucidate the cardiac-specific impacts of RAF1[S257L] on sarcomere structure, contractile behavior, $Ca^{2+}$ handling, and intracellular signaling. To this end, two iPSC lines were derived from two different individuals carrying the recurrent *RAF1* c.770 C > T (p.Ser257Leu) variant, of which one was gene-corrected, and three independent control iPSC lines were used (Fig. 1a). Our study provided the following important findings: (i) Monogenic RAF1[S257L]-related ultrastructural abnormalities at the sarcomere, including shortening of the I-band, are associated with an isoform shift of titin from a longer (N2BA) to a shorter isoform (N2B). This phenotype is associated with disrupted sarcomere structures, which were also found in a heart muscle biopsy of one NS patient. (ii) Elevated BNP levels, which is a common indicator of HCM, were observed in our model. (iii) Sarcoplasmic reticulum $Ca^{2+}$-ATPase (SERCA2A) and L-type calcium channels (LTCC), two main regulators of intracellular calcium transients, were downregulated, resulting in a shift in the SERCA2/PLN ratio. Calcium transient amplitudes in RAF1[S257L]-CMs were also smaller consistent with reduced amounts of free calcium that cycle between the sarcoplasmic reticulum and the cytosol. These results, therefore, suggest a role in the aberrant contraction frequencies and contractile tensions in our 3D model. (iv) Functional analysis of the RAF1[S257L]–BCTs revealed increased spontaneous contraction frequencies, myocardial thickening, lower contractile tensions, aberrant

contraction kinetics, and a significant decrease in the energy efficiency of the working myocardium. (v) Increased MAPK, p38, and increased YAP signaling events in CBs contribute to altered functional behavior and calcium transients. (vi) Treatment with the MEK inhibitor rescued most of the observed hypertrophic phenotype caused by RAF1 GoF, which was validated using the gene-corrected isogenic control. Importantly, the critical structural findings of the in vitro models were consistent with the results of a myocardial biopsy of one of the NS individuals.

## Results

**Generation of iPSC lines.** To model the RAF1-induced HCM based on patient-derived iPSCs, two NS individuals with severe HCM, heterozygous for the pathogenic *RAF1* c.770 C > T variant (p.Ser257Leu; RAF1[S257L] hereafter), were recruited for this study (Fig. 1a). iPSC lines were generated from both individuals using integration-free methods (Figs. S1 and S2) as previously described[17]. Furthermore, an isogenic gene-corrected iPSC line from one of these individuals was generated *via* CRISPR/Cas9 technology (Fig. S2h). In parallel, control iPSC lines from three healthy individuals (iPSC-WT1–3) were used and characterized (Fig. S3)[18]. The colonies of the iPSC-WT and iPSC-RAF1[S257L] lines were positive for different pluripotency markers, including alkaline phosphatase (Fig. S1a, b and S3a, b), OCT4, TRA-1-60, and SSEA4 (Figs. S1c, S2a–c, and S3c). Sequencing verified the presence of the heterozygous *RAF1* c.770 C > T variant in the iPSC-RAF1[S257L] lines as well as the wild-type status of the control iPSC lines (Figs. S1d, S2d, and S3d) and karyotyping confirmed the stable chromosomal integrity of the iPSC lines (Figs. S1e, S2e, and S3e). Spontaneous EB formation and spontaneous undirected differentiation confirmed the tri-lineage differentiation potential of each line (Figs. S1f, g, S2f, g, and S3f, g).

**Cardiac differentiation of iPSC lines.** Human iPSC lines were differentiated into CMs in 3D aggregates by temporal modulation of canonical WNT signaling as described previously (Fig. 1b, c)[19,20]. Analyses showed that the cells were negative for the presence of pluripotency markers at mRNA and protein levels and positive for cardiac markers (Fig. 1d–f and S5a). At d40 of differentiation, CBs were more than 96% cTNT-positive and 96% MLC2v-positive, as analyzed by flow cytometry (Fig. 1d). Immunofluorescence (IF) of cryosections of CBs (d47) revealed homogenous populations of cTNT- and α-actinin-positive cells in the center as well as in the periphery of the CBs (Fig. 1e, f).

Cardiac hypertrophy is defined as the enlargement of the CMs. iPSC-derived CMs were tested for their inability to proliferate in response to external stimuli. To this goal, d40 CBs were dissociated, and single CMs were seeded on coverslips. Thereafter, CMs were fixed and assessed for the presence of the mitotic marker phospho-Ser10-histone 3 (p-H3), confirming their negative p-H3 staining at d47 (Fig. 2a). As a positive control, proliferative iPSCs were treated for 12 h with 100 nM nocodazole (NC) to be arrested in mitosis (Fig. 2b and S4a, b). Cell cycle analysis indicated that NC-treated human iPSCs were mainly captured at the G2/M phase and stained positive for p-H3 (Fig. 2b). Moreover, dissociated CMs at d40 were treated with 100 μM L-phenylephrine (PE), an α-adrenergic agonist, for 7 days. These cells remained p-H3 negative but increased in size (Fig. 2c, d), which is consistent with PE's pro-hypertrophic activity. Collectively, these data indicate that in our experimental system employed iPSC-derived CMs could increase in size in response to hypertrophic stimuli but did not undergo proliferation (hyperplasia), providing a suitable in vitro model to study the molecular mechanism implicated in RAF1-driven HCM.

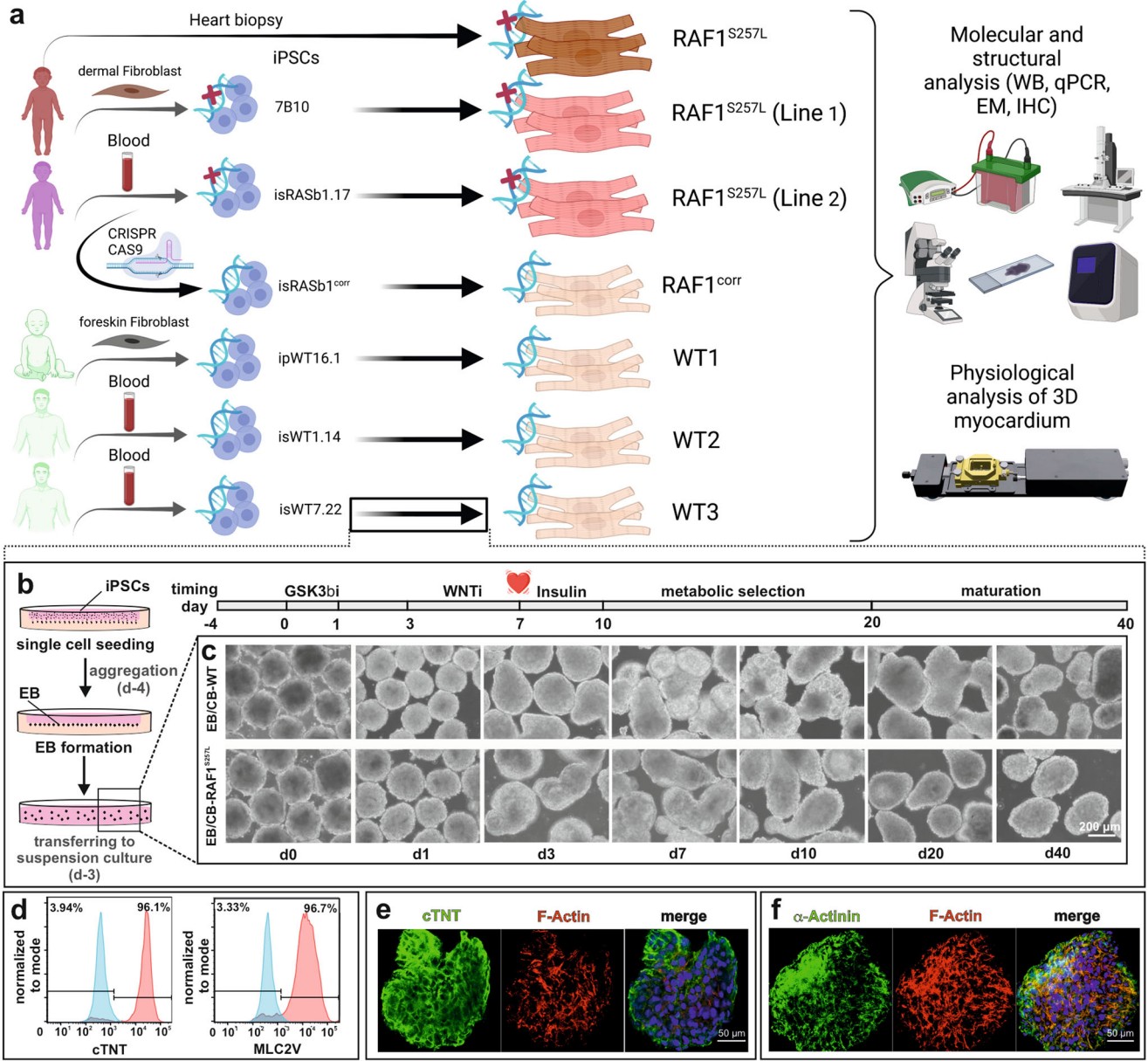

**Fig. 1 Study overview and details of 3D cardiac differentiation of iPSCs with WNT signaling modulation. a** Summary of the donor cells and the iPSC lines together with an overview of different analysis approaches used in this study. EM electron microscopy, IHC immunohistochemistry, qPCR quantitative PCR, WB western blot. **b** Schematic overview of embryoid body (EB) formation using agarose microwells combined with the stages and timelines of EBs differentiation to cardiac bodies (CBs). **c** Light microscopic pictures of EBs/CBs during the course of cardiac differentiation and metabolic selection. **d** Exemplary histograms of flow cytometric analysis of dissociated CBs displayed efficient cardiac differentiation towards ventricular CMs by analysis of MLC2V and cTNT positive cells (RAF1$^{S257L}$, line 1). Isotype controls are depicted in light gray. **e, f** Immunofluorescence staining of a representative CB for cTNT and α-actinin proteins (RAF1$^{S257L}$).

**Signaling events in RAF1$^{S257L}$-CBs.** Serine$^{257}$ (S257) is located in close proximity to the negative regulatory phosphorylation site S259 within the conserved region 2 (CR2) of RAF1 (Fig. 3a). The latter provides a docking site for 14-3-3 proteins, thereby stabilizing RAF1 in its auto-inhibited state[4,21]. To determine the impact of the S257L substitution on the S259 phosphorylation status in cells with endogenous expression of the kinase, we immunoprecipitated total RAF1 protein with an anti-RAF1 antibody from lysates of undifferentiated iPSCs-WT and iPSCs-RAF1$^{S257L}$ (line1). Immunoblot analysis with anti-RAF1 and anti-p-RAF1$^{S259}$ revealed up to 44% reduction in the levels of RAF1$^{S257L}$ phosphorylated protein, compared to WT-RAF1 at S259 (Figs. 3b and S8a). Therefore, due to the heterozygous status

of the mutation in the model system, it can be assumed that the majority of the mutant RAF1 protein remains unphosphorylated, accounting for almost 50% of total RAF1. This observation demonstrates a reduced ability of RAF1$^{S257L}$ to be subjected to the 14-3-3 inhibitory control at physiological conditions.

Next, we investigated the activity of selected RAS/RAF-dependent signaling in WT and RAF1$^{S257L}$-CBs (line1) in the presence and absence of 0.1 μM MEK inhibitor PD0325901 (MEKi; Fig. 3c). In untreated CBs, the PI3K-AKT-S6K-mTORC-AKT and RAF1-ASK1-JNK signaling axes did not show remarkable differences between RAF$^{S257L}$- and WT-CBs (Fig. 3d, e and S8b). However, increased levels of p-ERK1/2 and p-p38, and decreased levels of p-YAP in RAF1$^{S257L}$-CBs were

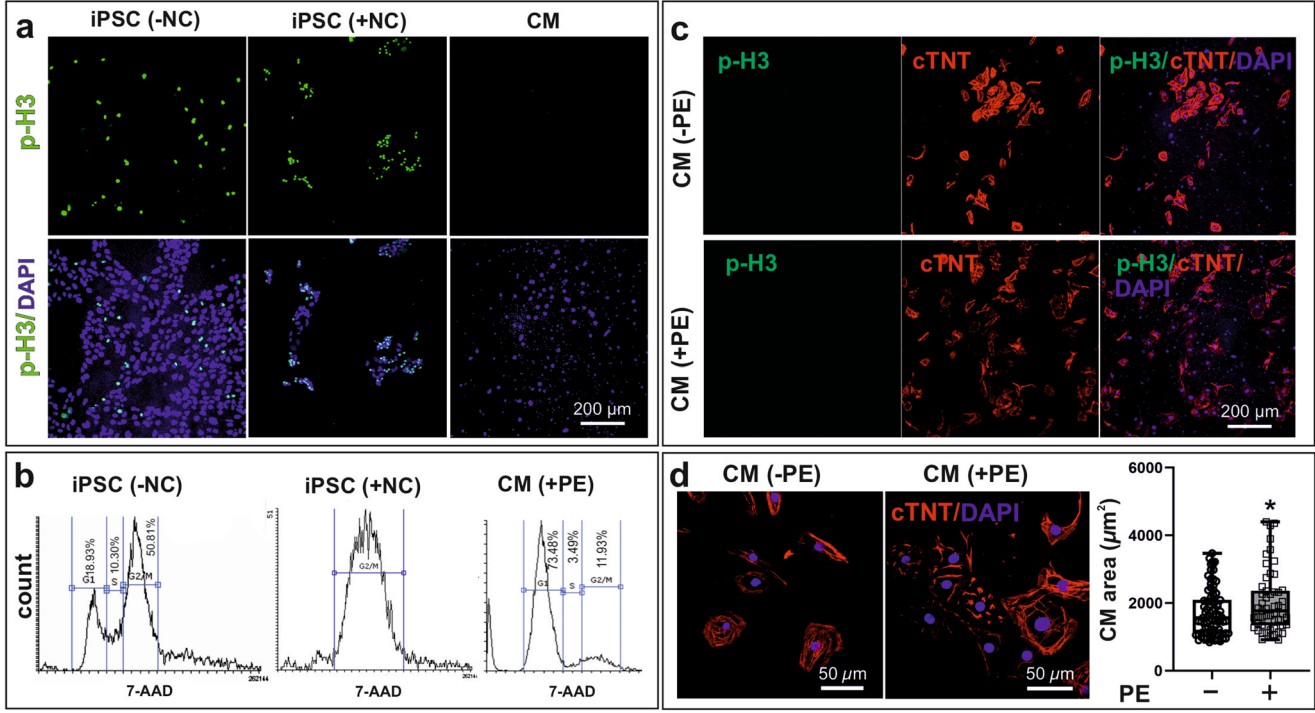

**Fig. 2 Hypertrophy versus hyperplasia in iPSC-derived CMs. a** Illustration of mitotic cells stained with the mitotic marker phospho-Ser10-histone 3 (p-H3) in iPSCs and dissociated CBs at d40. As a positive control, proliferative human iPSCs were treated for 12 h with 100 nM nocodazole (NC) to be arrested in mitosis. CMs show no proliferative behavior as compared to iPSCs, which were arrested in mitosis by NC treatment. **b** Cell cycle analysis of iPSC untreated and treated with NC as well as CMs treated with 100 μM L-phenylephrine (PE). Cell cycle analysis indicates that NC-treated iPSCs are mainly captured in the G2/M phase and stained positive for p-H3. Where PE-treated CMs are arrested in G1. **c** Dissociated CMs at d40 were treated with PE for 7 days. Both treated and untreated CMs remain p-H3 negative. **d** The cell surface area of the stained CMs with the cardiac marker of cTNT (−PE and +PE) were quantified with Image J software, which indicates the increased cell size in response to PE's pro-hypertrophic activity. *$P < 0.05$, unpaired 2-tail t-test. $n = 2$, biological replicates. CM cardiomyocytes, cTNT cardiac troponin T, iPSC induced pluripotent stem cells, PE phenylephrine, p-H3 phospho-histone 3, NC Nocodazole.

documented. The p-ERK1/2 and p-p38 levels were significantly reduced upon treatment with MEKi, while a significant increase in the level of p-YAP was observed (Fig. 3d, e and S5b, c, S8d; line 2). Furthermore, we examined the impact of the signaling signature of CRISPR-corrected RAF1$^{corr}$-CBs *vs.* its mother clone (RAF1$^{S257L}$-CBs) and found the opposing pattern of phosphorylation of the former pathways (Fig. 3f), which highlights the explicit impact of the RAF1 point mutation at Ser257 on the observed signaling patterns.

Quantitative real-time PCR (qPCR) analysis showed significant upregulation of *NPPB* in RAF1$^{S257L}$-CBs, which was partially reverted in the presence of MEKi (Fig. 3g and S5d). The gene *NPPB* encodes for BNP (brain natriuretic peptide), a well-known clinical biomarker for heart failure, and is upregulated during hypertrophy due to a return to a fetal-like gene expression program. Enhanced expression of *NPPB* has also been shown in vitro to be mediated by p-ERK under stretch-induced conditions[22]. Next, we examined the levels of secreted pro-BNP in the medium of the cultured CB variants. Interestingly, the results indicated that RAF1$^{S257L}$-CBs secrete more than 12- and 30-fold amounts of BNP in their medium compared to MEKi-treated and gene-corrected control CBs, respectively (Fig. 3h). Notably, the isogenic control elucidated an even lower amount of BNP than MEKi-treated cells, which may indicate the activity of the parallel pathways beside RAF1-MAPK in the regulation of BNP levels, e.g., p-RAF1-MST2-YAP.

**RAF1$^{S257L}$ alters stretch-shortening of sarcomere.** Familial non-syndromic HCM caused by mutations in sarcomeric proteins is known to affect sarcomere architecture[23]. To address the

question of whether the sarcomere architecture is also altered in RAF1$^{S257L}$-associated HCM, CBs at d40 were dissociated and single CMs were cultured for 7 days on coverslips. RAF1$^{S257L}$-CMs showed less oriented and more disarrayed myofilaments as compared to WT-CMs (Fig. 4a), which was confirmed by immunohistochemistry (IHC) and electron microscopy (EM). The former observation was in the same line of evidence with a previous study of RAF1$^{S257L}$-CMs (d20) that were developed in a 2D culture system[8]. In particular, RAF1$^{S257L}$-CBs revealed shortened or more contracted I-band regions and thickened Z-line pattern (Fig. 4b). Left ventricular cardiac tissue (CT) was available for one of the NS individuals with RAF1$^{S257L}$, and staining of desmin and cardiac troponin I confirmed a disorganized sarcomere structure with shortened I-bands, as seen on iPSC-derived CMs and CBs along with a thickened Z-line (Fig. 4c, d). We measured the cell surface area of the single cells in overview pictures of CB-EM. RAF1$^{S257L}$-CBs exhibited a significant increase in cell size (on average 135%) compare with WT-CBs (Fig. 4e). Notably, RAF1 was localized and condensed near the sarcomere structures in RAF1$^{S257L}$-CMs, while in WT-CMs RAF1 expression was more cytoplasmic (Fig. 4f). The sarcomere structures of the cardiac cells were quantified using Sota software (Fig. 4f) developed for quantifying sarcomeres[24]. The organization values range from 0 to 0.3, where 0 represents disorganized and 0.3 represents organized sarcomeres. The quantification data confirm that RAF1$^{S257L}$-CMs have less organized and aligned sarcomere structures (Fig. 4g).

To examine the influence of the dysregulated MAPK signaling on the observed phenotype, we treated the RAF1$^{S257L}$-CBs with PD0325901 (0.2 μM) at the early stages of development (d12)

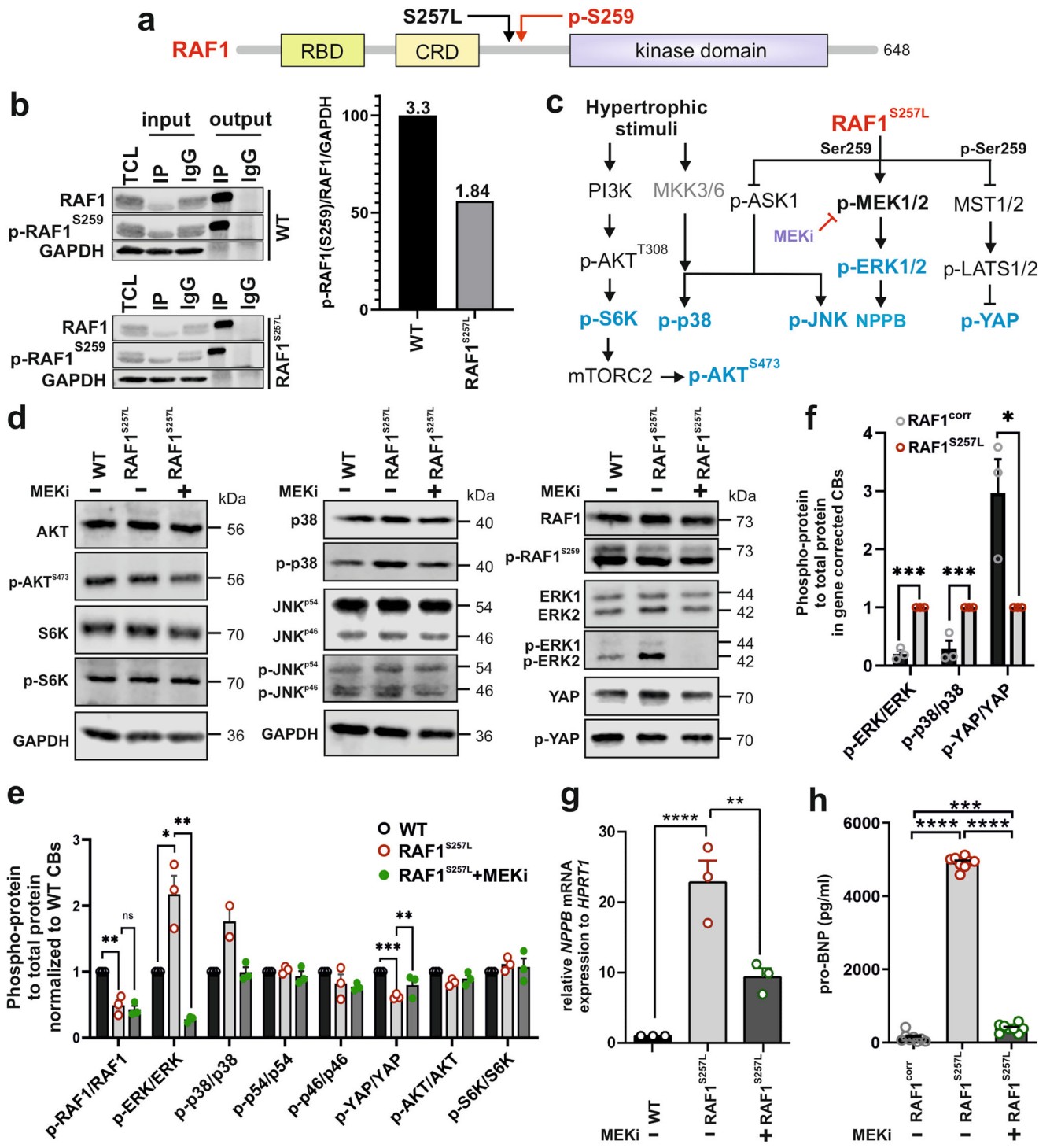

until d40. Remarkably, ultrastructure analysis revealed that MEKi treatment restored the I-band width around the Z-line in the RAF1^S257L-CBs (Fig. 4h and S5h). To confirm the observed I-band shortening, we generated RAF1^S257L- and WT-BCTs (Fig. S6), prepared cryosections of these BCTs, and dissected a part of giant sarcomeric protein titin by immunostaining the PEVK domain of titin to label the I-band region and α-actinin staining to indicate Z-lines (Fig. 5a). PEVK domains of titin and α-actinin were strikingly co-localized in RAF1^S257L-BCTs, while separated from each other in WT-BCTs (Fig. 5b), suggesting a major shortening of the I-band region with dislocation of the titin

PEVK region in the RAF1^S257L-BCTs. Area histograms of the selected sub-images (Fig. 5b; boxed and magnified) were created to determine the distance between the two PEVK segments and quantify the average distances of more than 50 different Z-lines per sample. These abnormalities of the overlapped peaks corresponding to the two PEVK domains relative to α-actinin were remarkably restored and comparable to WT-BCTs when RAF1^S257L-BCTs were treated with MEKi (Fig. 5c). The significant separation of the distance between the two adjacent PEVK segments upon MEKi treatment strongly suggests a functional normalization of the sarcomere structures in

**Fig. 3 The effect of the RAF1$^{S257L}$ variant on the activity of selected effector kinases downstream of RAF1. a** Domain organization of RAF1 kinase with the typical functional domains, including the RAS-binding domain (RBD), the cysteine-rich domain (CRD), and the kinase domain. The adjacent sites of the S257L variant and the inhibitory S259 phosphorylation (p-S259) are highlighted. **b** Immunoprecipitation and quantification of total and p-RAF1$^{S259}$ in WT and RAF1$^{S257L}$ iPSCs (line 1). Total RAF1 was immunoprecipitated from lysates of WT and RAF1$^{S257L}$ iPSCs using an anti-RAF1 specific antibody. IgG was applied as an isotype control. Immunoblotting was carried out using anti-RAF1 and anti-p-RAF1$^{S259}$ antibodies. For quantification, signal intensities of p-RAF1$^{S259}$ were divided by those for total RAF1. GAPDH was used as a loading control. TCL total cell lysate, IP immunoprecipitation, IgG immunoglobulin G. **c** Schematic diagram summarizing the signaling molecules investigated downstream of hypertrophic stimuli and RAF1. Proteins marked in blue letters were investigated at the protein level by immunoblotting. **d** Representative immunoblots of p-AKT *vs.* AKT, p-S6K *vs.* S6K, p-RAF1$^{259}$ *vs.* RAF1, p-ERK1/2 *vs.* ERK1/2, p-YAP *vs.* YAP, p-p38 *vs.* p38, and p-JNK *vs.* JNK using cell lysates from WT- and RAF1$^{S257L}$-CBs (d24). **e** Phospho-protein *vs.* total protein ratio quantification as shown in (**d**). *$P < 0.05$, **$P < 0.01$, ***$P < 0.001$, ****$P < 0.0001$, unpaired 2-tail *t*-test. #$P < 0.05$, ##$P < 0.01$, unpaired 1-tail *t*-test. $n = 3$, biological replicates, except for p-38/p38 ($n = 2$). **f** Phospho-protein *vs.* total protein ratio quantification of western blot results for selected pathways in RAF1$^{S257L}$-CBs *vs.* gene-corrected line, RAF1$^{corr}$-CBs (d24). *$P < 0.05$, **$P < 0.01$, ***$P < 0.001$, unpaired 2-tail *t*-test. $n = 3$, biological replicates. **g** qPCR analysis of *NPPB* transcription levels. **$P < 0.01$, ****$P < 0.0001$, unpaired 2-tail *t*-test, $n = 3$, biological replicates. **h** ELISA analysis of pro-BNP levels released in the cell culture supernatant of CB's (pg/ml). ***$P < 0.001$, ****$P < 0.0001$, unpaired 2-tail *t*-test. $n = 8$. Error bars for (**e–h**): + SEM.

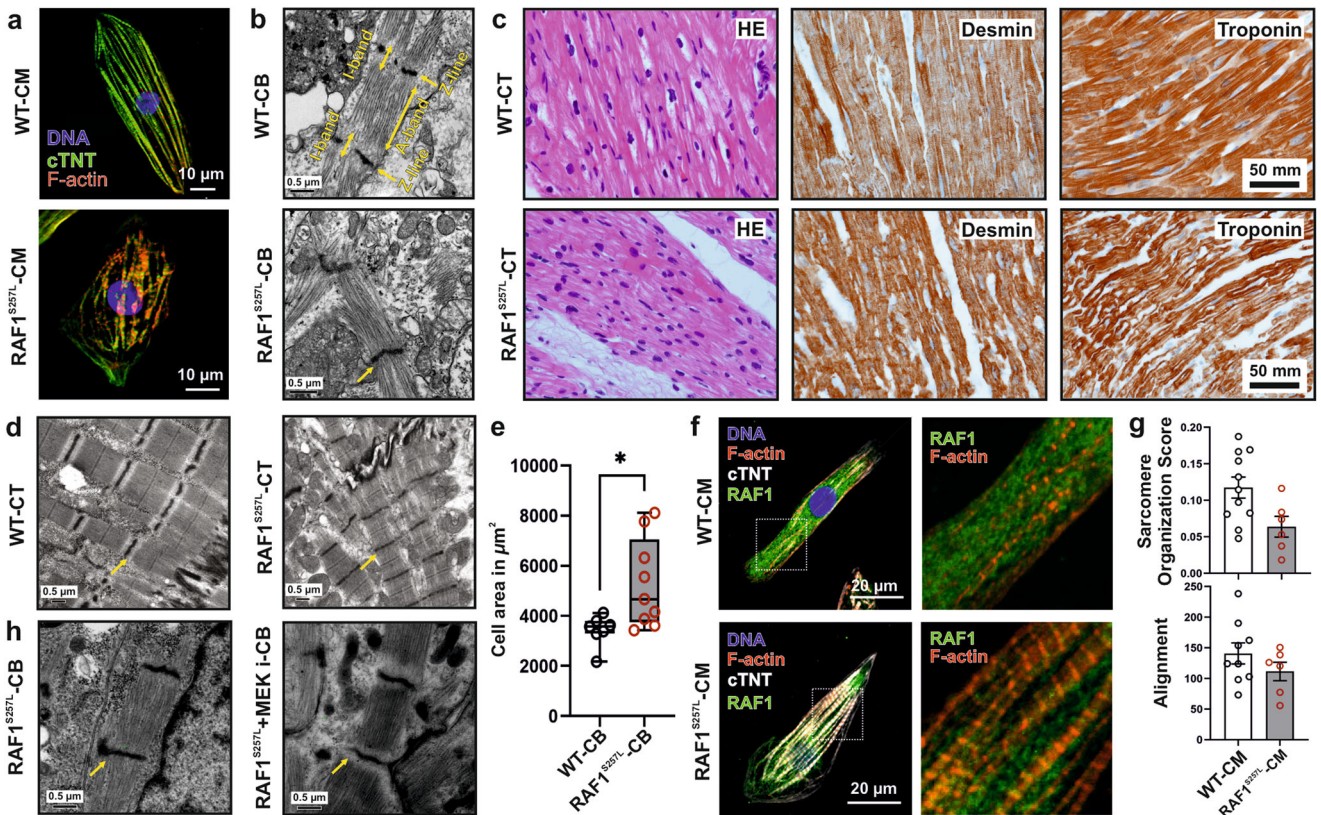

**Fig. 4 Aberrant RAF1$^{S257L}$ activity impairs the cytoarchitecture of human iPSC-derived cardiomyocytes. a** Dissociated WT- and RAF1$^{S257L}$-CBs (line 1) were seeded on Geltrex-coated coverslips for 7 days and stained for cTNT and F-actin (Scale bar, 10 μm). **b** Representative EM images from RAF1$^{S257L}$-CBs revealed stronger myofibrillar disarray accompanied by shortened I-bands and a thickened Z-line pattern as compared to WT-CBs. **c** IHC analysis of RAF1$^{S257L}$ cardiac tissues (CTs) from one of the NS individuals with *RAF1 c.770 C>T* variant for desmin and troponin showed myofilament disarray. **d** Representative EM images of the same RAF1$^{S257L}$-CTs as in C exhibited shortened I-bands and a thickened Z-line pattern consistent with RAF1$^{S257L}$-CBs in (**b**). **e** Quantification of the cell size area of the CB-EM pictures with image J software. *$P < 0.05$, unpaired 2-tailed *t*-test. $n = 2$. **f** Representative ICC images of RAF1$^{S257L}$ and WT-CMs at d90 post-differentiation showed RAF1 co-localization with cTNT and F-actin at the sarcomere (Scale bar, 10 μm). **g** The quantifications of the organization and alignments of the sarcomeres in RAF1$^{S257L}$- and WT-CMs at d90 post-differentiation stained with anti-cTNT antibody, Fig. 3f, with Sota software. $n = 2$. **h** EM images of RAF1$^{S257L}$-CBs (d40) treated with 0.2 μM MEKi from d12 of differentiation.

RAF1$^{S257L}$-BCT (Fig. 5d). Moreover, by measuring the average distance between α-actinin signals as the Z-line marker, a decrease in the sarcomere lengths was observed for the RAF1$^{S257L}$-BCTs compared to WT-BCTs (Fig. 5e). We further determined a significant increase in the mRNA levels of the predominant longer and more compliant (*N2BA*) titin isoform towards the shorter and stiffer isoform (*N2B*) in RAF1$^{S257L}$ BCTs (Fig. 5f).

Collectively, the data demonstrate that RAF1 GoF promotes I-band shortening and reduces the flexibility of the spring elements of the titin I-band region.

**Aberrant expression of sarcomeric regulators.** The impaired sarcomere organization of RAF1$^{S257L}$-CMs prompted us to quantitatively analyze the expression of key sarcomeric

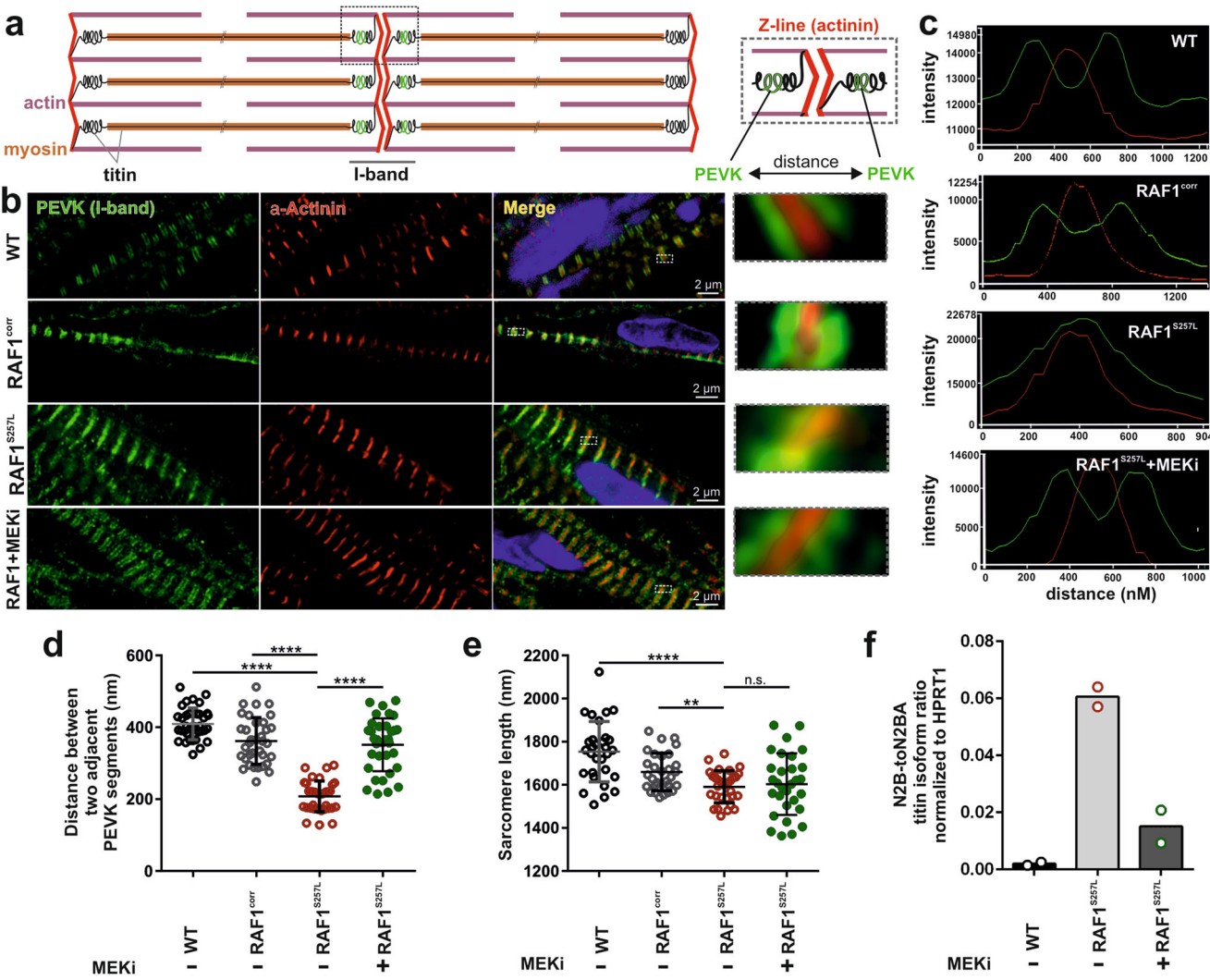

**Fig. 5 Hyperactive RAF1$^{S257L}$ triggers a shorter I-band phenotype. a** Schematic view of the sarcomere organization. **b** IHC analysis of 8-μm cryosections bioartificial cardiac tissues (BCTs) from WT, gene-corrected, RAF1$^{S257L}$, and RAF1$^{S257L}$-treated with MEK inhibitor with PEVK segment of titin's I-band, α-actinin as the Z-line marker, and DAPI for DNA staining. **c** A histogram of selected boxes on G was imported based on the intensity and overlaps of emitted fluorescent lights using the Zeiss LSM 880 Airyscan confocal microscope software. **d** Averaged distance (nm) between two adjacent PEVK segments was measured and statistically evaluated for more than 50 different sarcomere units for each condition ($n = 35$). ****$P < 0.0001$, unpaired 2-tailed $t$-test. **e** Averaged sarcomere length (nm) was measured for more than 50 different sarcomere units for each condition by measuring the distance between two parallel Z-lines (α-actinin) and statistically evaluated ($n = 30$). ****$P < 0.0001$, unpaired 2-tailed $t$-test. **f** qPCR analysis of the ratio of the *N2B-to-N2BA* titin isoforms expression levels in cardiac bodies ($n = 2$). *$P < 0.05$, **$P < 0.01$, unpaired 2-tailed $t$-test.

components, including troponins, myosins, and actin-related proteins. In comparison to WT-CBs, RAF1$^{S257L}$-CBs strikingly exhibited higher levels of *MYH7* and *MYL2*, but lower levels of *TTN*, *MYH6*, *MYL7*, and *α-SMA* (Fig. 6a). We analyzed the *MYH7*-to-*MYH6* ratio in CBs at two different maturation stages (d24 and d47). Both, immature (d24) and more mature CBs (d47) displayed a significant increase in the *MYH7*-to-*MYH6* ratio (Fig. 6b and S5e). Notably, MEKi treatment partially reversed the *MYH7*-to-*MYH6* ratio in d47, but not d24 in RAF1$^{S257L}$-CBs.

**Reduced Ca$^{2+}$ transients RAF1$^{S257L}$-CMs.** Next, we investigated a possible dysfunctional calcium handling of RAF1$^{S257L}$-CBs, which is a central feature of HCM[10]. We first analyzed the expression of the components that regulate intracellular calcium cycling in WT and RAF1$^{S257L}$-CBs at d47. These components include ryanodine receptor type-2 (RyR2), SERCA2A, phospholamban (PLN), and the calcium voltage-gated channel alpha

(CACNA) subunits 1C and 1D (LTCCs). *PLN*, *SERCA2A*, *RYR*, and *CACNA1C* mRNA expression was downregulated in RAF1$^{S257L}$-CBs (Fig. 6c). Furthermore, *SERCA2A* and *CACNA1D* expression was significantly reduced in RAF1$^{S257L}$-CBs treated with MEKi (Fig. 6c). Remarkably, the decrease of the *SERCA2-to-PLN* ratio in RAF1$^{S257L}$-CBs was reversed to WT levels upon MEKi treatment in d47 CBs (Fig. 6d and S5f).

The main function of SERCA2 is transporting calcium from the cytoplasm to the sarcoplasmic reticulum and it is negatively regulated by PLN. Phosphorylation of PLN at Ser[16] by protein kinase A inhibits PLN activity. SERCA2a, PLN, and p-PLN$^{Ser16}$ were reduced in RAF1$^{S257L}$ compared to WT-CBs (Fig. 6e and S8c). The SERCA2/active PLN ratio, which was calculated at protein levels by measuring the ratio of SERCA2 to PLN/p-PLN$^{Ser16}$, was also significantly reduced in RAF1$^{S257L}$-CBs (Fig. 6f). The changes in SERCA2a and PLN expression and the ratio of PLN to p-PLN$^{Ser16}$ were consistent with aberrant calcium handling properties, a characteristic of maladaptive hypertrophy[10,25].

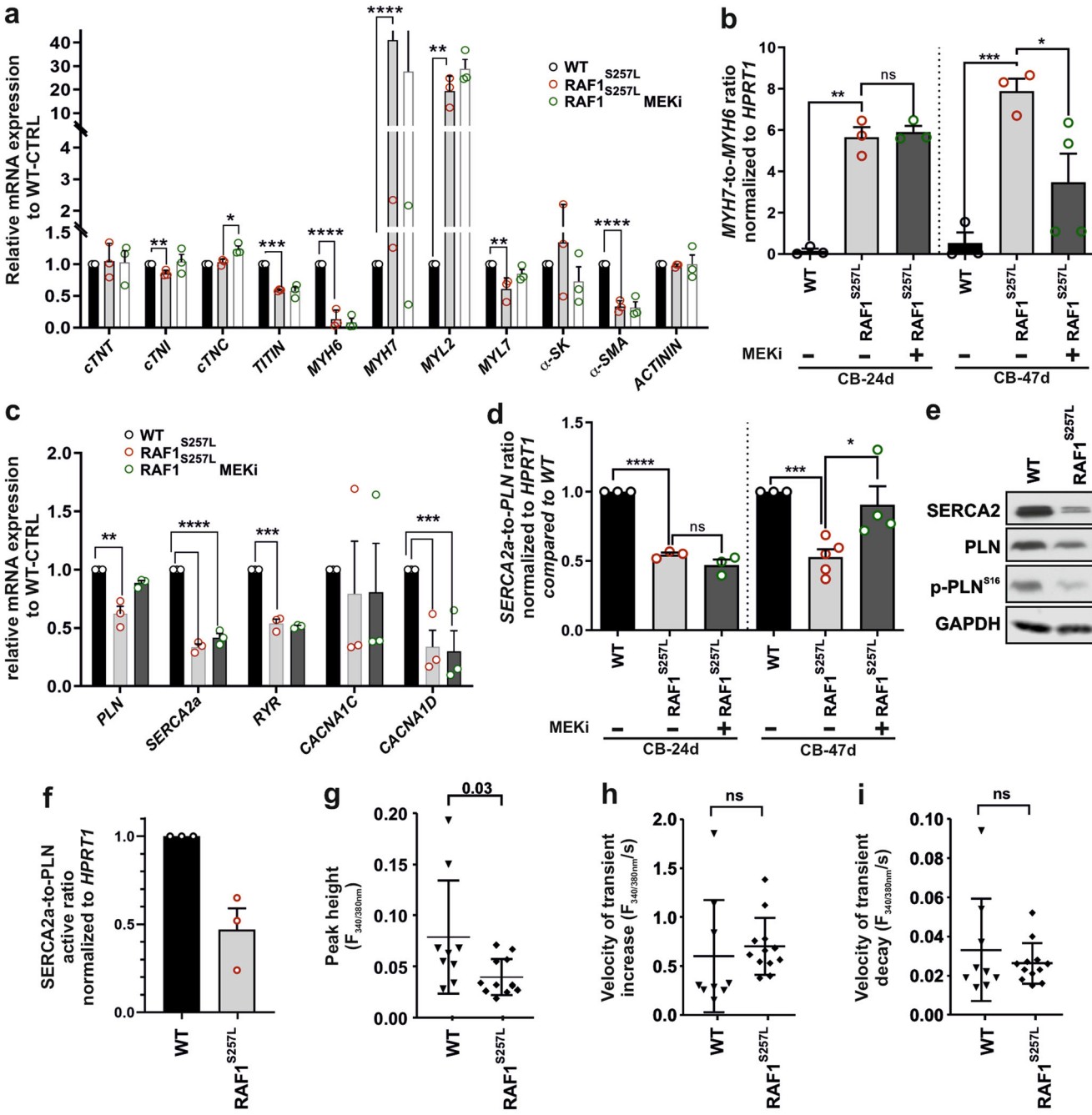

**Fig. 6 Abnormal expression of proteins involved in sarcomere function and calcium handling.** The experiments in A-F were conducted in cardiac bodies (CBs) of WT, RAF1[S257L] (line 1), and RAF1[S257L]-treated with 0.2 μM MEKi from d12 of differentiation. The data are averaged from three independent experiments in biological triplicates. *$P < 0.05$, **$P < 0.01$, ***$P < 0.001$, ****$P < 0.0001$, unpaired 2-tail $t$-test. $n = 3$, biological replicates. **a** qPCR analysis of mRNAs related to sarcomere proteins. **b** MYH7-to-MYH6 ratio. **c** qPCR analysis of mRNAs related to the regulation of calcium transients. **d** SERCA2a-to-PLN ratio obtained from qPCR data. **e** Immunoblot analysis of SERCA2, PLN, and p-PLN[Ser16] in CBs. **f** The ratio of SERCA2 to PLN was calculated by measuring the ratio of SERCA2 to PLN/p-PLN[Ser16] in western blotting. Error bars for **a–d**, **f**: +SEM. **g-i** Ca[2+] transients were measured in Fura2-loaded dissociated CMs in 2D and expressed as the ratio of fluorescence emission at 340 and 380 nm. Bar graphs display the peak height of Ca[2+] transients (**g**) and the velocities of cytosolic Ca[2+] increase (**h**) and decrease (**i**). Each data point represents the average of 10 transients obtained from a single CM. Nine wild-type CMs ($n = 9$) and twelve RAF1[S257L]-CMs ($n = 12$) were analyzed in total.

Therefore, we measured intracellular calcium transients by seeding CMs (d47) on Geltrex-coated coverslips and loaded the cells with the calcium indicator Fura-2. The calcium transient was significantly decreased in RAF1[S257L]-CMs as compared to WT-CMs (Fig. 6g). Despite the reduced rate of calcium transients, the kinetics of cytosolic calcium rise and decrease in RAF1[S257L]-CBs were not

significantly different from WT-CBs (Fig. 6h, i). Collectively, these data suggest that the RAF1[S257L] signaling might modulate the contractile cardiac function by increasing the ratio of PLN to SERCA2 at the molecular levels and therefore impairing the balance of intracellular calcium cycling. However, the exact mechanism needs to be investigated.

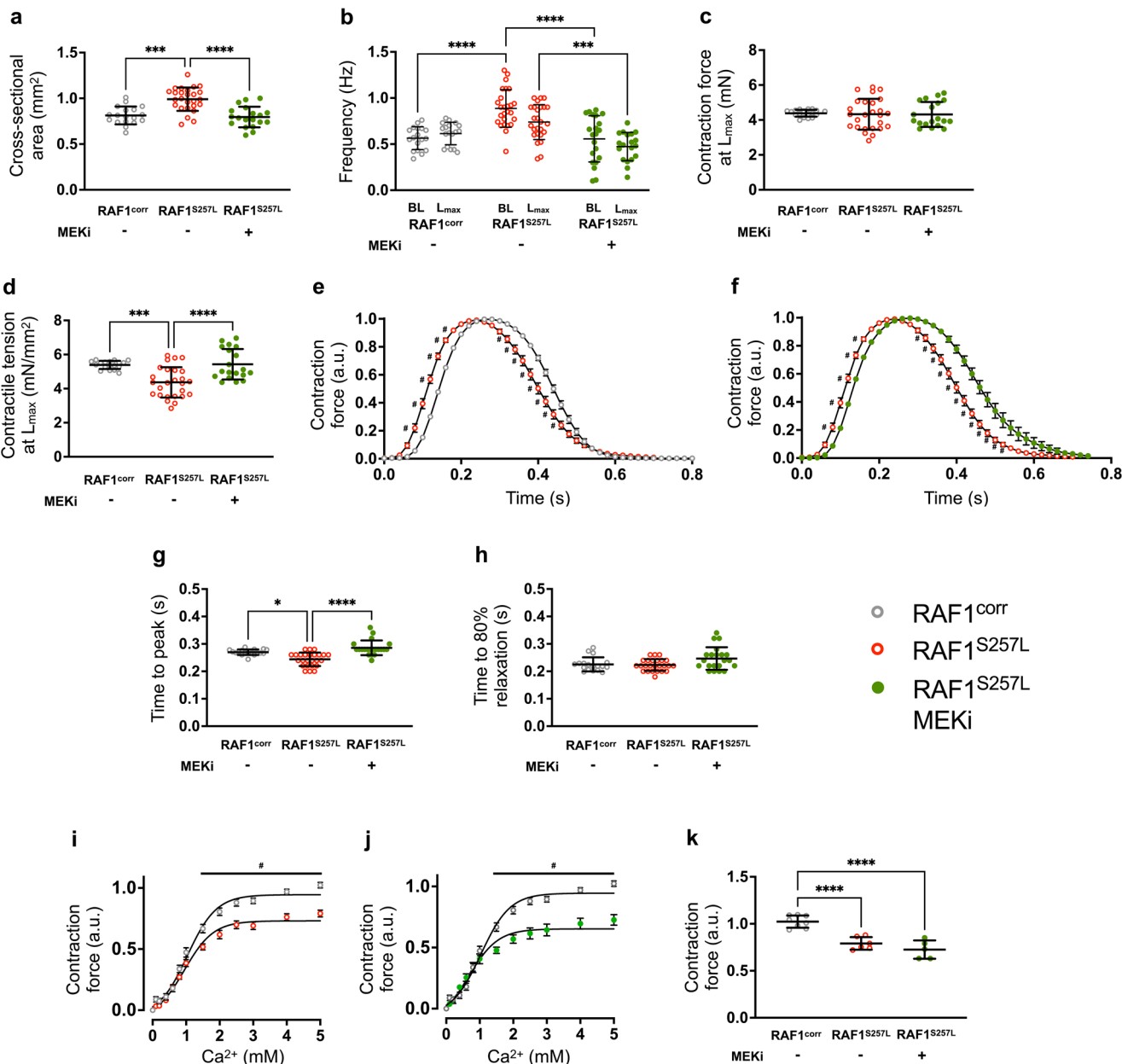

**Fig. 7 Aberrant physiology of RAF1^{S257L}-BCTs and their response to inhibition of MEK.** Measurements were conducted on days 27-28 of 3D tissue culture using isogenic RAF1^{corr}, RAF1^{S257L}, and RAF1^{S257L}-BCTs + MEKi. **a** Quantification of cross-sectional areas of BCTs derived from the different genotypes and treatments. **b** Spontaneous beating frequencies at baseline (BL) and at the preload step, where the maximum contraction force was recorded for each BCT, i.e., at L_{max}. **c** Maximum contraction forces of BCTs at L_{max} at 1 Hz electrical pacing. **d** Maximum contractile tensions based on the cross-sectional areas of BCTs. Contraction kinetics at L_{max} compared between **e** RAF1^{corr} *vs.* RAF1^{S257L} and **f** RAF1^{S257L} *vs.* RAF1^{S257L} + MEKi. Quantification of **g** time to peak of the contraction and **h** time to 80% relaxation at L_{max}. Responses of BCT contractions to increasing calcium concentrations (range: 0.1–5 mM) in 5 mM glucose conditions normalized to contraction forces measured in high glucose (25 mM) medium compared between **i** RAF1^{corr} *vs.* RAF1^{S257} and **j** RAF1^{corr} *vs.* RAF1^{S257L} + MEKi. **k** Quantification of normalized maximum contraction forces at 5 mM calcium under low glucose (5 mM) levels. **a–h**: n = 14-26 individual tissue samples per group. **i–k** n = 5–8 individual samples per group. Depending on the presence of normally distributed values, one-way ANOVA or Kruskal–Wallis test was applied. Error bars for **a–d** and **g**, **h**: ±SD; Error bars for **e–j** ±SEM. *P < 0.05, #P < 0.01, ***P < 0.001, ****P < 0.0001.

## The negative impact of RAF1^{S257L} on contraction physiology.

Next, we analyzed the effect of RAF1^{S257L} on microtissue physiology using a multimodal bioreactor system to generate, cultivate, stimulate, and characterize BCTs (Fig. S6). Compared to gene-corrected controls, RAF1^{S257L}-BCTs showed a significant increase in the cross-sectional area (CSA) and significantly increased beating frequency, with both parameters positively influenced by MEKi (Fig. 7a, b and S7a–d). Although no

differences in absolute maximal contractile forces were observable in the RAF1^{S257L} myocardium, contractile tension was significantly decreased (Figs. 7c, d and S7e). Again, treatment with the MEK inhibitor resulted in normalization to control values. Furthermore, the significantly altered contraction kinetics of RAF1^{S257L} myocardium, i.e., the shape of contraction, was adjusted by MEKi (Fig. 7e, f). The contraction velocity, indicated by the time to peak contraction (TTP), was significantly faster in

the mutant myocardium, and MEKi effectively slowed this parameter down to control levels (Fig. 7g). The time to 80% relaxation was not significantly affected by either the mutation or MEKi, although a slight tendency to slow was observed in the latter group (Fig. 7h). We investigated the sensitivity of RAF1$^{S257L}$ myocardium to calcium in a titration experiment in Tyrode's solution containing standard glucose concentration, i.e., 5 mM (Fig. 7i). Although no differences in responses to increasing calcium concentrations were observed between the groups, the contraction forces in the mutant myocardium did not reach the values measured in normal cultivation medium containing 25 mM glucose. This phenomenon, which points toward impaired energy efficiency, was not altered by MEKi (Fig. 7j, k).

## Discussion

The in vitro cellular reprogramming, differentiation, and tissue engineering of patient-derived samples reproducibly generated CBs and BCTs as human 3D disease models to investigate the molecular events contributing to cardiac dysfunction in RAF1-related NS.

Patient-derived iPSCs and their isogenic gene-corrected controls are an invaluable source for CMs for human 3D in vitro disease models to investigate the molecular events and structural anomalies contributing to myocardial dysfunction in RAF1-related NS.

Human 3D cardiac models allow the extensive molecular and functional analysis of cardiac disease mechanisms to a level beyond 2D systems and animal models.

The advantages of the generation of patient-specific models and the possibility of using gene editing techniques like CRISPR/Cas9 allow for a better understanding of the links between specific genetic mutations and pathologies and provide versatile options for therapeutic testing on these patient-specific disease models. This genetically determined human disorder is caused by the aberrantly enhanced function of the RAF1 kinase. We characterized the CBs' and BCTs' cytoskeletal and sarcomere ultrastructures by super-resolution and EM imaging, calcium handling, contractility, and intracellular signal transduction. These complementary approaches identified reduced *MYH6* abundance over *MYH7*, elevated expression of *NPPB* and secretion of pro-BNP, decreased *SERCA2/PLN* ratio, reduced force generation accompanied by a reduced rate of intracellular calcium transients, increased levels of p-ERK1/2, p-p38, and attenuation of p-YAP, as signatures of RAF1$^{S257L}$-CMs. Most remarkably, RAF1$^{S257L}$-CBs and -BCTs as well as heart biopsy samples from the RAF1$^{S257L}$ individuals revealed common ultrastructural features, namely shortened I-bands. The alterations in titin isoform balance (N2B-to-N2BA shift) and shortened I-bands of RAF1$^{S257L}$-CBs/-BCTs were attenuated by treatment with the MEK inhibitor PD0325901. MEK inhibitors have been used clinically as a therapeutic option for NS patients in different concentrations resulting in significant improvement in cardiac status and repression of cardiac hypertrophy in addition to normalization of pro-BNP levels in these patients[26–28]. Collectively, our results suggest RAF1$^{S257L}$-mediated activation of the MAPK pathway produces an abnormal cardiac phenotype involving structural and physiological aspects, that can be rescued with MEKi.

**RAF1$^{S257L}$ signaling in human CM.** CMs with heterozygous RAF1$^{S257L}$ exhibited an approximately 50% reduced inhibitory phosphorylation of RAF1 at S259, which is consistent with previous reports[1,2,8]. The best-studied RAF1 function is the activation of the MEK1/2-ERK1/2 pathway, which regulates the activity of a wide range of signaling molecules in the cytoplasm and nucleus. Accordingly, higher p-ERK1/2 levels were detected in

RAF1$^{S257L}$-CMs compared to WT-CMs, also consistent with previous studies[6,8], and, p-ERK1/2 levels were remarkably reduced in CBs upon MEKi treatment. In addition to its crucial role in normal heart development, the MAPK pathway may also act as a central signaling node for many factors stimulating adaptive and maladaptive hypertrophy[29,30]. However, the detailed mechanisms involving aberrant RAF1$^{S257L}$-MEK1/2-ERK1/2 signal transduction to induce hypertrophy remain unclear. This phenomenon may be rooted in the regulation of cardiac specific-transcription activators or/and direct/indirect transcriptional modulations of cardiac components *via* ERK1/2 regulation (nuclear substrates) as well as direct modulation of cardiac function by affecting contractile machinery (cytoplasmic/sarcomeric substrates; Fig. 8). Accordingly, in RAF1$^{S257L}$-CBs compared to WT- and MEKi-treated CBs, we detected disruption in the expression pattern of contractile machinery proteins, including a transient shift from MYH6 to MYH7, in addition to calcium transient regulators. The observed changes in the expression for these genes likely result from transcriptional activation of the hypertrophic-responsive gene promoters by GATA4, AP1, MEF2, NFAT, and NFκB, as described by previous studies[31–33]. Future investigations employing chromatin immunoprecipitation of these transcription factors from isolated nuclei of RAF1$^{S257L}$-CBs combined with proteomic approaches may clarify the transcriptional regulation by the MAPK pathway.

Activated RAF1 binds to MST2 (also called STK3) and inhibits the MST1/2-LATS1/2 pathway[34]. As a consequence, YAP translocates into the nucleus, associates with TEAD to serve as a transcriptional co-activator, and regulates transcription of the mitogenic factors, including CTGF, NOTCH2, and c-MYC as well as miR-206. The MST1/2-LATS1/2-YAP axis is critical for heart development, growth, regeneration, and physiology[35]. It regulates proliferation in the neonatal heart and growth and survival in the adult heart[36–38]. The YAP-miR206-FOXP1 axis regulates CM hypertrophy and survival by modulating the expression patterns of important genes, e.g., *MYH7* and *NPPB*[39]. We demonstrate here that the reduced inactivating phosphorylation of YAP in RAF1$^{S257L}$ was restored upon MEKi treatment (Fig. 3d–f), which is likely explained by the enhanced RAF1$^{S257L}$ kinase activity along with a switch of RAF1 binding from MST1/2 to MEK1/2 (Fig. 3c)[34]. Two different groups have reported that the transcription of *MYH7* and *NPPB* genes appears to be under the control of YAP-TEAD1[24,40]. Akerberg et al. analyzed the chromatin occupancy by different transcription factors GATA4, NKX2-5, MEF2A, MEF2C, SRF, TBX5, and TEAD1, using CHIP-Seq data. They identified TEAD1-binding peaks for *MYH7* and *NPPB* genes[40]. Similarly, we observed increased expression of *MYH7* and *NPPB* in RAF1$^{S257L}$-CBs, which correlates with our signaling data and may be due to increased active YAP levels in RAF1$^{S257L}$-CBs. Highly elevated levels of the *NPPB* gene product pro-BNP were detected in the supernatant of RAF1$^{S257L}$-CBs, considerably above the critical clinical thresholds defined for the likelihood of a heart failure state in patients (Fig. 3h)[41]. This validates the observed signaling impact of hyperactive RAF1$^{S257L}$ signaling on the fetal gene expression programs, directing the cells towards a heart failure condition. Considerably, we inhibited the MEK-MAPK axis in cardiac cells without targeting RAF1 directly, therefore, we expected that only the MAPK-dependent phenotype is rescued upon MEKi treatment. However, we observed that parallel pathways downstream of RAF1 are also reverted by MEKi treatments such as MST2/YAP.

We propose that MEK inhibition abrogates the negative feedback phosphorylation of RAF1 by ERK, and thus restores RAF1 membrane localization and activity[42].

Other MAPKs besides ERK1/2, such as p38, JNK, and ERK5, appear to be involved in cardiac development, function, and also

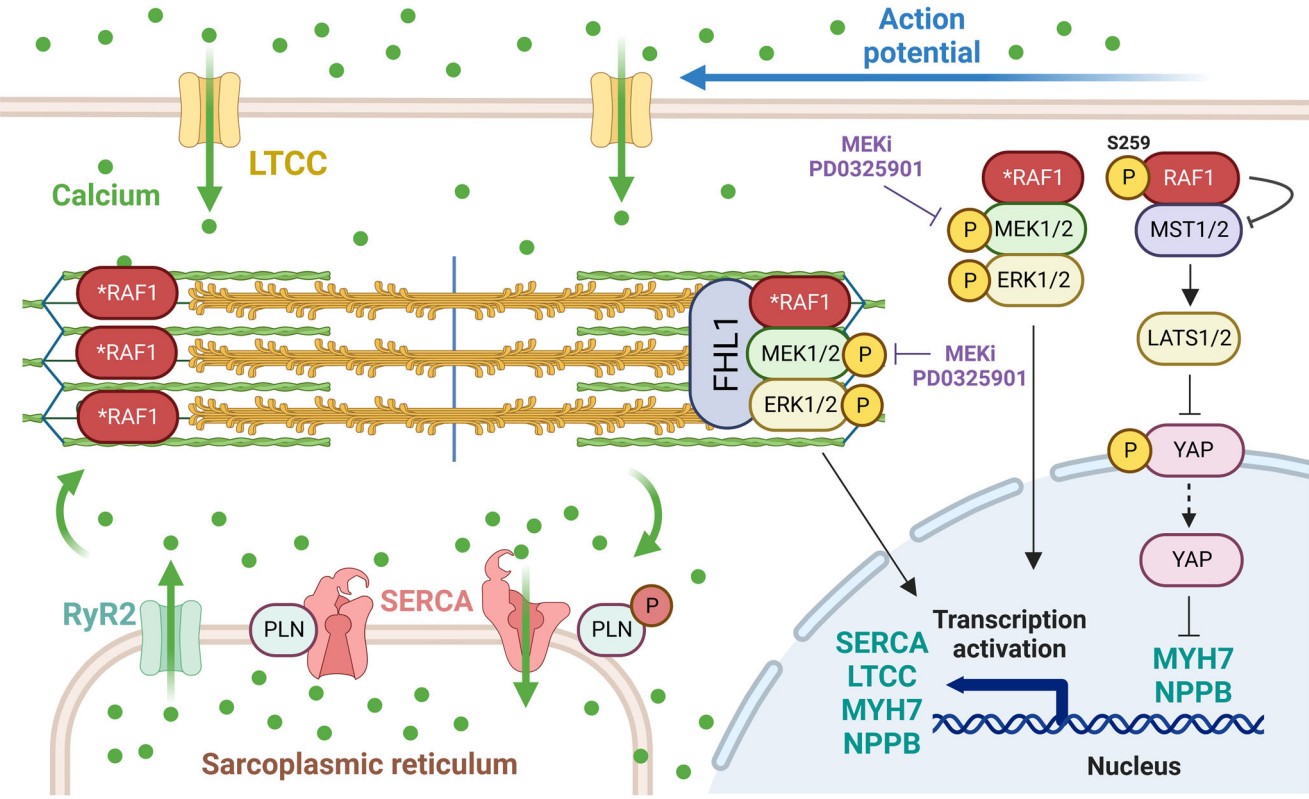

**Fig. 8 A proposed model of both *RAF1*-dependent cardiac signaling pathways, coupling calcium transients and contraction, and RAF1[S257L]-enhanced impairment of cardiac contraction, force generation, and calcium transients.** Due to critical ultrastructural defects in the sarcomere that impair the necessary flexibility components, which are crucial stress sensors, and localize relevant MAPK signaling proteins, the RAF1 mutant cardiomyocytes exhibit more MAPK signaling events. The above-mentioned events accompanied by the perturbation of the transcriptional profile of cardiomyocytes resulting from the hyperactivating mutation in RAF1 and further enhanced downstream signaling axes contribute to aberrant calcium transients through altered levels of SERCA and LTCC calcium transporters and PLN as suppressor of SERCA. While affected MYH6 and MYH7 compositions along with other variables contribute towards impaired contractility and force generation, the disease state is further fueled by massively increased pro-hypertrophic signals *via* NPPB/BNP.

the progression of myocardial disease[8,29]. Our data indicated an increase in p38 phosphorylation, but not JNK, in RAF1[S257L]-CBs. Aberrant p38 phosphorylation was reverted to (near) normal upon MEKi treatment. The molecular mechanism that underlies p38 activation *via* RAF1[S257L] in cardiac cells is unclear. We propose that elevated levels of p-p38 in RAF1[S257L] cells may be a compensatory response of cells to reduce the effects of the sustained ERK activity towards an unknown ERK-dependent or ERK-independent positive feedback regulation of p38[43,44].

HCM has been reported to correlate with ERK5 activation (reviewed in [30]). (i) CM-specific ERK5 KO mice showed a lower hypertrophic response and induction of apoptosis during pressure overload; (ii) stimulation of ERK5 was associated with eccentric and detrimental hypertrophy in an animal model of long-term intermittent cardiac hypoxia; (iii) ERK5 activation is promoted by angiotensin II hypertrophic detrimental stimulus. Jaffré et al. argue for cardiac-specific RAF1 signaling to ERK5 and calcineurin-NFAT in HCM although they did not observe differences in ERK5 phosphorylation[8]. They instead inhibited the MEK5/ERK5 pathway with BIX02189, an inhibitor of both MEK5 (IC50 = 1.5 nmol/L) and ERK5 (IC50 = 59 nmol/L) showed that treatment of RAF1[S257L]-CMs with BIX02189 significantly reduced cell surface area.

We did not pursue this further for the following reasons: (i) We also analyzed ERK5 phosphorylation and observed hardly any pattern of p-ERK5 with RAF1[S257L] with and without MEKi as well as RAF1[corr]. (ii) ERK5 activation could be EGFR-dependent but RAS-independent, as well as independent of the RAF-MEK-

ERK axis[45]. (iii) The small molecule inhibitor BIX02189 also has an inhibitory effect on the TGFβ receptor (TGFβR) and SMAD pathway[46]. Given this important point and the known importance of SMAD1/5 activity as a driver of pathogenic cardiac hypertrophy, it may be possible that the observation by Jaffré et al. with BIX02189 could be due to the inhibition of the TGFβR/SMAD pathway[8].

**Altered cardiac calcium handling in RAF1[S257L] CM.** One characteristic of maladaptive hypertrophy is an abnormal calcium handling that affects myocardial contractility[10,25]. In the myocardium, calcium-induced calcium release is essential for excitation-contraction coupling[47]. Depolarization of the plasma membrane through action potential results in LTCC and calcium influx. RYR2 is highly sensitive to small changes in calcium concentration and becomes activated upon local calcium influx, which then facilitates the release of $Ca^{2+}$ from the sarcoplasmic reticulum into the cytosol. Binding of $Ca^{2+}$ to troponin C causes tropomyosin translocation, which exposes actin filaments for binding to myosin heads, cross-bridge formation, and triggers contraction (systole). During diastole, $Ca^{2+}$ is transported into the SR by SERCA2A.

At the transcriptional level in RAF1[S257L]-CMs, we observed the downregulation of two main regulators of intracellular calcium transients, *SERCA2a* and *LTCC*, as well as changes in the *SERCA2/PLN* ratio. MEKi treatment of RAF1[S257L]-CMs restored the *SERCA2/PLN* ratio and significantly downregulated *NPPB* to

levels similar to that of WT-CMs. This suggests that SERCA2A/PLN ratio and *NPPB* expression may be under direct or indirect transcriptional control of ERK1/2 (Fig. 8). Previous studies have shown that the RAS-MAPK pathway regulates *SERCA2A*, *PLN*, and *LTCC* expression that is downregulated during hypertrophy and heart failure[33,48,49]. Consistently, an increase in BNP, which is known as an HCM biomarker, leads to a decrease in *SERCA2A* expression[50].

In addition to the observed transcriptional alterations of calcium regulators, RAF1[S257L]-CMs also had reduced intracellular $Ca^{2+}$ transients. The influence of hyper-activation of RAS signaling on expression and regulation of SERCA2 and PLN-mediated calcium handling and its role in diastolic dysfunction in HCM has been previously reported in mouse models[51]. A lower SERCA2/PLN ratio in RAF1[S257L]-CMs may thereby cause a delay in $Ca^{2+}$ re-entry to the SR *via* inhibition of SERCA by PLN and thus changes the kinetics of calcium transients and consequently decreases the capacity of cardiac contractility[52]. Therefore, a decreased SERCA2a/PLN ratio could be considered as the stressor to induce the HCM phenotype[49,53]. In a similar manner, iPSC-derived CMs with GoF variants in BRAF and MRAS displayed changes in intracellular $Ca^{2+}$ transient[16,54]. In contrast, a recent study that compared the idiopathic HCM and RAF1[S257L]-associated HCM has shown that CMs from idiopathic cases, but not RAF1[S257L]-CMs, exhibit significant alterations in calcium handling[7].

**Unusual contractile units in RAF1[S257L]-CM**. RAF1[S257L]-CMs revealed a disorganized sarcomere structure (Fig. 4a-d). A remarkable finding in the present study is the atypical I-bands, in both RAF1[S257L]-CMs as well as the heart biopsy samples from the corresponding individual with the heterozygous *RAF1[S257L]* variant. We observed this phenotype in several independent experiments, and it was completely re-established upon MEK inhibition. This was also reported by a previous study on the RAF1[S257L]-CMs[8]. Additionally, our highly structured cardiac tissue allowed us to have a closer look at the ultrastructure of BCTs. IHC of the selected region of I-band, the PEVK domains of titin, and Z-line (α-actinin) indicated that two adjacent PEVK regions in RAF1[S257L]-BCTs overlapped (in green) on the Z-line (in red), whereas in WT, the Z-line was surrounded by two distinct and well separated PEVK regions (Fig. 5b). Treatment of BCTs with 0.1 μM MEKi resolved these abnormalities (Fig. 5b, c). The I-band segment of titin acts as a molecular spring that develops tension when sarcomeres are stretched, representing a regulatory node that integrates and perhaps coordinates diverse signaling events[55]. The four-and-a-half LIM domain 1 protein (FHL-1) has been shown to bind to titin at the elastic N2B region and to enhance cardiac MAPK signaling by directly interacting as a scaffold protein with RAF1, MEK2, and ERK2[56]. In our experiments, *FHL1* mRNA expression was up-regulated in RAF1[S257L]-CMs (Fig. S5g). Additionally, we observed that RAF1 was predominantly localized alongside the sarcomeres in RAF1[S257L]-CMs (Fig. 4f). Interestingly, the N2B region of titin has been identified as a substrate of ERK2, and phosphorylation by ERK2 reduces the stiffness of titin[57,58]. We propose that RAF1[S257L] hyperactivates the MEK1/2-ERK1/2 pathway, which is most likely localized alongside the sarcomeres, *via* FHL1, and enhances titin phosphorylation at its N2B region. To what extent these events may result in altered sarcomere distensibility and contributes to the cardiac abnormalities observed with RAF1[S257L] remains to be determined (Fig. 8).

Consistent with our observed phenotype of the atypical I-band region in both the cardiac in vitro model and biopsy of the NS-RAF1 individual, studied the ultrastructure of the myocardial biopsies from cardiomyopathies[59]. They showed the EM pictures of the HCM patient that also lacks the I-band, whereas in other cases such as infundibular stenosis and aortic regurgitation, the I-band was visible. Another study investigated myocardium obtained from 22 patients with asymmetric septum hypertrophy. They also showed disarrayed sarcomere structures and reported some abnormalities in Z-bands. Although the authors did not mention directly that the I-band is not visible, they indicated some abnormalities in Z-bands. Notably, in their electron microscopy micrographs, we cannot see I-bands[60]. Additionally, Maron et al.[61] presented some EM pictures from hypertrophied cardiac tissue with the same phenotypes as their previous studies of the HCM specimens that altogether showed no I-band in their micrographs[61].

**RAF1[S257L] effects on myocardial function**. Expectedly, the physiology and contractile behavior of cardiac tissue was affected by the molecular alterations, structural abnormalities, and lower intracellular $Ca^{2+}$ transients in RAF1[S257L]-CMs[62]. In our gene-corrected model of miniaturized 3D myocardium, we observed different phenotypes and the effects of MEKi. In addition to the increase in tissue size, as determined by a 22 % larger CSA, our detailed analysis of myocardial physiology revealed that the spontaneous contraction frequency of RAF1[S257L]-BCTs was increased by 57% which is an indicator of the HCM state in the RAF1[S257L]-BCTs and reduced contractile tension by 19%. These three parameters were fully rescued by MEKi. In addition, the higher velocities of contraction kinetics in RAF1[S257L]-BCTs, possibly caused by a stiffer PEVK region in titin, were also positively modulated by MEKi. Interestingly, this treatment did not affect the potential loss in energy efficiency in the mutant myocardium. The phenomenon of impaired energy homeostasis is also known from other RASopathies, e.g., Costello syndrome, in which affected individuals also suffer from HCM[63].

In addition, certain aspects of myocardial physiology were significantly altered by RAF1[S257L]. Here, we have shown that deviant contractile properties such as spontaneous contraction frequency and contractile tension were positively influenced by the inhibition of MEK. However, the impaired energy inefficiency caused by RAF1[S257L] was not affected by MEKi. Future studies will be needed to address this unaffected feature of RAF1[S257L] myocardium. One promising approach may be to target the AMPK pathway, as has been described to modulate altered bioenergetics and reduce left ventricular hypertrophy in a mouse model of Costello syndrome[64]. In addition, it will be important to determine whether it is possible to restore normal physiology after the RAF1[S257L]-associated phenotype has manifested strongly. From a clinical perspective, systemic pharmacological intervention would usually be started postnatally when the hearts of affected individuals have already undergone developmental changes that significantly impair cardiac function.

The link between the RAF1-MAPK signaling pathway and contractile behavior in the myocardium is unclear. One explanation for the altered contractile behavior of RAF1[S257L]-CMs may be the perturbed *MYH6* (α-MYH)-to-*MYH7* (β-MYH) switch due to the aberrantly activated MAPK signaling (Figs. 6b, 8, and S5e). ERK1/2 are known to phosphorylate the cardiac-specific transcription factor GATA4 at S105 and enhance its transcriptional DNA-binding activity. GATA4 is critical for the expression of structural and cardiac hypertrophy response genes, such as *NPPB*, *MYH7*, *TNNI3* (troponin I), and *ACTA1* (α-skeletal Actin)[31,32,65–67]. The expression of MYH isoforms changes during normal heart development and disease progression. During human heart development, there is a shift from α-MHC to β-MHC. However, in end-stage HCM, there is generally

even a higher transition in the ratio of β-MHC to α-MHC. An in vitro assay showed that α-MHC is faster and has a higher ATPase activity, and generates fewer forces than β-MYH[68]. We assume that cardiac-specific transcription factors mediate the upregulation of myosin heavy chain isoforms by RAF-MAPK. The ATP hydrolyzing capacity of the two myosin heavy chain paralogs are dissimilar; α-MYH has a 3-fold higher ATPase activity and generates more force than β-MYH, which affects the velocity of myofibril shortening and, consequently, contraction[69]. Reduced *MYH6* levels have also been reported in human heart failure[70]. Therefore, we propose that RAF1^S257L-CMs exhibited higher levels of *MYH7*, leading to less absolute force generation and contractile tension. Alternatively, an increased PLN-to-SERCA2 ratio and reduced intracellular calcium transients may affect intracellular calcium concentrations and cross-bridge cycling kinetics. In addition to the *MYH6*-to-*MYH7* switch, further factors including SERCA2/PLN ratio, titin phosphorylation by ERK1/2, disorganized sarcomere structures, and changes in length/shape of the flexible I-band region of titin, might also affect force generation and elastic properties of the myocardium. Cardiac contraction-relaxation processes are multifactorial and complementary analyses are required for more clear-cut conclusions.

Although the analyzed RAF1^S257L mutation is not directly affecting sarcomeric proteins as most of the HCM-causing mutations, we observed its drastic impact on the sarcomere organization in CMs in patient myocardial biopsies and in vitro. Next to the potential impact of the dysfunctional titin composition as seen in RAF1^S257L-CMs, resulting in a loss of flexibility, the observed changes in contraction kinetics might be explained by an increased sliding velocity of thin filaments of the sarcomere, as caused by mutations in sarcomeric proteins in HCM[71–73]. The aspect that in HCM the "sliding velocity" is increased might be useful to explain the contraction shapes and kinetics (Fig. 7e–g). Eschenhagen and Carrier recently reported that another parameter of the sarcomere function is the unloaded sliding velocity of thin filaments on immobilized S1 myosin. Several studies suggest that HCM mutations are associated with increased sliding velocity and dilated cardiomyopathy (DCM) with decreased sliding velocity[74].

Increased cell size is more indicative of HCM rather than an eccentric hypertrophic phenotype generally associated with DCM. We have observed an increase in the cell size of CBs (Fig. 4E) and CSAs of BCTs (Fig. 7a).

A very important point is that the majority of DCM cases are due to abnormalities of titin and sarcomere at the A-band region[75,76]. This is again a different phenotype from that observed in our disease model, with HCM abnormalities mainly affecting the I-band region of titin and the sarcomere which is a hallmark of HCM (Fig. 4). Hinson et al. confirm this critical difference using cardiac microtissues differentiated from human iPSCs with patient-specific A-band truncation variants of titin leading to pathogenic DCM phenotypes[77]. On the other hand, the results in other studies using HCM-associated human iPSCs are consistent with the results of our study, not only in terms of the affected I-band region, but also in terms of other key HCM features such as sarcomere disorganization, cellular enlargements, and the central role of the hyperactivation of ERK1/2 signaling as a contributing factor to the development of HCM in models based on familial cardiomyopathy[10,13,78,79].

In addition, we found evidence for dysfunctional myocardial energy metabolism as another element contributing to the phenotype caused by RAF1^S257L. Similar abnormalities were recently described in an iPSC-based 3D model of sarcomere-linked HCM and were associated with myocardial dysfunction[11]. These authors also observed higher spontaneous contraction

frequencies, impaired contraction forces, as well as a switch from energy-hungry αMHC to energy-efficient ßMHC. The latter has been reported for a variety of human iPSC-based HCM models, but not for DCM models[74]. Furthermore, a recent study reported a relatively low MYH7 expression in human iPSC-derived CBs, which may explain our observation of pronounced changes in expression levels of this protein in RAF1^S257L-CBs in comparison to the wild-types and isogenic control lines used[80]. Systolic dysfunction is more pronounced in DCM, whereas diastolic dysfunction is more pronounced in HCM[78]. We did not find impaired absolute contraction forces in BCTs, but an altered contractile tension (Fig. 7c, d).

Another hallmark of HCM has elevated BNP levels (NPPB), which we also detected in RAF1^S257L-derived cardiac cells[78]. Impaired calcium handling has also been reported in iPSC-derived CM of familial HCM[10]. There are opposing hypotheses on the effects of mutations on hyper- or hypocontractility[81]. However, hypocontractility and hypercontractility are described alternatively in the pathogenesis of HCM, depending on the gene- or mutation-specific effects, the cardiac model used, and the experimental design. In previous studies using an iPSC-derived model of HCM, both hyper- and hypocontractility have been reported[82]. Remarkably, most of the relevant data come from mutations of sarcomeric proteins, e.g., MYH7, not in signaling components.

The pathogenesis we observed related to RAF1^S257L, such as disarrayed sarcomere structures, abnormalities in the I-bands, and shifts in titin isoforms, suggests that we may have changes in contractile behavior due to these defects (Fig. 7). However, we cannot say whether we had hypo- or hypercontractility by looking only at the maximum force development data in Fig. 7c. Therefore, we prefer to leave this question open so that further studies with a more similar cardiac system can answer this question.

Despite the discovery of numerous mutations in genes underlying HCM and DCM, our understanding of the patho-mechanisms leading from mutation to phenotype is still incomplete. Reasons for this are not only the diversity of mutations causing similar disease patterns, the incomplete and highly variable penetrance of HCM and DCM but also the fact that mouse models only partially recapitulate the human phenotype (reviewed in Eschenhagen and Carrier)[74]. Despite the value of mouse models in the possibility to study the development of the effects of heart disease in the whole organism, there are some remarkable aspects of human models that cannot be studied in mice. For example, the resting heart rate of mice is ten times higher than that of humans. In mice, myosin heavy chain 6 (MYH6; faster isoform) is highly expressed in the ventricle, whereas in humans it is MYH7 (slower isoform). The development of the heart, the contribution of ion channels, and, accordingly, the electrical properties differ greatly between humans and mice[83]. In any case, the experience with mouse models raises the question of the extent to which they truly reflect human disease and provides a strong argument to study HCM and DCM in human iPSC-derived CMs. The electrophysiology of the heart and cardiac cells, as well as their frequency and force of contraction, have a major impact on several significant physiological characteristics. Furthermore, RASopathy-associated HCM typically shows its greatest manifestation dynamics in the peri- or early postnatal period, suggesting a developmental phenotype. The maturity of our human iPSC-based 3D model seems to be sufficient to phenocopy many of the features that play a crucial role in the pathophysiology and progression of the disease and is a valuable tool to analyze the possibilities, but also the limitations, of MEK inhibition in terms of phenotype rescue in vitro.

Collectively, we demonstrated new aspects of RAF1 function in human iPSC-derived CMs, which resemble the observed in vivo

phenotype from the corresponding individuals, especially changes in the ultra-structure of the sarcomeres. The S257L variant in RAF1[S257L]-CMs modulates RAF1-dependent either signaling networking, fetal gene program, increase in cell size, beating frequency, contraction, calcium transients, the sarcomeric components, and structures. By applying the MEKi, we confirmed that reduced p-p38, BNP levels, I-band length, SERCA2-to-PLN ratio, contractile tension, and increased p-YAP, N2B-to-N2BA, beating frequency are regulated directly or indirectly downstream of RAF1[S257L]-MEK-ERK axis. However, the contraction force generation was not significantly increased upon the treatment and its link to the RAF1[S257L] needs to be addressed. We believe future studies with further advanced models of myocardium will uncover more precisely the physiological output of hypertrophic RAF1 variant(s). The next generation of 3D cardiac models with emphasis on the improvement of maturity of these cardiac models in addition to enhancement of limitations such as the absence of vasculature and blood circulation in these models would be remarkable in driving the field steps closer to mimicking humans in vivo conditions.

## Materials and methods

**Generation, cultivation, and gene-correction of iPSCs.** Blood samples and dermal fibroblasts were obtained with institutional ethics approvals (Justus-Liebig-University Giessen, Germany: AZ258/16, Otto von Guericke University Medical Center Magdeburg: 173/14; University Medical Center Göttingen: 10/9/15) and under informed consent of the parents from two unrelated individuals with NS carrying the heterozygous substitution c.770 C > T in exon 7 of RAF1.

Primary human cells were reprogrammed using either episomal reprogramming vectors Epi5[TM] [17] (Thermo Fisher Scientific #15960) or Sendai virus system Cytotune 2.0 (Thermo Fisher Scientific #A16517). The resulting iPSCs were clonally picked and expanded in mTeSR (Stemcell Technologies) or StemMACS iPS-Brew (Miltenyi Biotech) to UMGi164-A clone 1 (7B10, here referred to as RAF1[S257L] line 1) from patient 1 and UMGi102-A clone 17 (isRASb1.17, here referred to as RAF1[S257L] line 2) from patient 2, respectively, and passaged using Versene (STEMCELL Technologies) at a ratio of 1:3 to 1:6, depending on cell density. Before their utilization for experiments, clonal iPSCs were subjected to detailed characterization including Sanger sequencing to confirm the presence of the variant, iPSC morphology, assessment of expression of pluripotency markers by RT-PCR, IF staining and flow cytometry, and chromosomal integrity. Furthermore, the elimination of persisting reprogramming vectors was confirmed by PCR and RT-PCR, respectively. Three unrelated wild-types (WT) iPSC lines, UMGi163-A clone 1 (ipWT16.1, here referred to as WT1)[84], UMGi014-C clone 14 (isWT1.14, here referred to as WT2), and UMGi020-B clone 22 (isWT7.22, here referred to as WT3)[18] were used. The CRISPR-corrected isogenic iPSC line was used as a control. Gene correction of the RAF[S257L] variant (c.770 C > T, heterozygous) in the iPSCs from patient 2 was performed using ribonucleoprotein (RNP)-based CRISPR/Cas9 by targeting exon 7 of the RAF1 gene, as previously described[85]. The guide RNA target sequence was (PAM in bold): 5′-TGGATGTCAACCTCTGCCTC **TGG**-3′. For homology-directed repair, a single-stranded oligonucleotide with 45-bp homology arms was used. After picking clones, successful gene-editing was identified by Sanger sequencing and the CRISPR-corrected isogenic cell line UMGi102-A-1 (isRASb1-corr) underwent the same detailed characterization as mentioned above.

**Human iPSC culture.** Undifferentiated iPSC lines were cultured in iPS-Brew XF (Miltenyi Biotec) for expansion and stock preparation with a daily medium exchange. Before cardiac differentiation, the medium was changed to murine embryonic feeder cell-conditioned medium (CCM+), to enhance induction of differentiation. CCM+ consists of DMEM F12 + Glutamax, 15% Knock-out Serum Replacement, 1% non-essential amino acids (all Gibco), 100 μM ß-mercaptoethanol (Sigma), and 100 ng/mL bFGF (PeproTech; CCM+/100). Cells were passaged every three to four days by dissociation with Accutase and seeded onto Geltrex-coated (0.5%, Life Technologies) plasticware at a density of $5 \times 10^4$ cells per cm$^2$ in CCM+, containing the ROCK inhibitor Y-27632 (10 μM, Selleckchem, #S1049). The ROCK inhibitor was eliminated from the CCM+ on the following days.

To produce murine embryonic fibroblast (MEF)-conditioned medium, gamma-irradiated MEFs were seeded in T175 flasks at a density of $6 \times 10^4$ per cm$^2$ in MEF medium consisting of DMEM high glucose (Gibco, 21969-035), 1× L-Glutamine (Gibco, 25030-024), 1× non-essential amino acids (Gibco, 11140050), and 10% fetal calf serum (Sigma, F7524). Starting from day one of MEF culture, the following medium composition was conditioned for 20–24 h and collected and frozen at −20 °C: DMEM/F12 (Gibco, 10565018) supplemented with 15% Knock-out serum replacement (Gibco, 10828028) and 1x non-essential amino acids, 5 ng/

mL bFGF (Peprotech, 100-18B-1MG) and 0.1 mM ß-mercaptoethanol (Sigma, M3148). After seven days of conditioning, the MEF culture was discarded, and the collected CM medium resulting in -CCM+ was pooled, filter-sterilized, and frozen. Prior to use for iPSC culture, the CCM+ was supplemented with an additional 100 ng/mL bFGF.

**Tri-lineage differentiation of human iPSCs.** To assess pluripotency iPSCs in vitro, we induced spontaneous differentiation of human iPSCs into all three germ layers. For this, iPSCs were detached from feeder layers using 0.4% (w/v) type IV collagenase and resuspended in a differentiation medium consisting of IMDM + GlutaMAX supplemented with 20% (v/v) fetal calf serum, 1 mL L-Glutamine, 0.1 mM 2-mercaptoethanol, and 1% non-essential amino acid stock (all Thermo Fisher Scientific). Cells were maintained for 7 days in suspension culture on 1% (w/v) agarose/IMDM coated 12-well plates to form 3D embryoid bodies (EBs). Subsequently, about 15–20 EBs were plated on 6-well plates coated with 0.1% (w/v) gelatine. After 24 days, EBs were harvested for qRT-PCR and replated for IF analyses, respectively.

**Cardiac differentiation of human iPSCs.** Cardiac differentiation was performed in 3D suspension culture after aggregate formation in agarose microwells (AMW) modified from the established protocol[19,20,86]. For differentiation in 3D suspension culture, AMW were generated from AggreWellTM400Ex plates (Stem Cell Technologies, #27840) containing 4700 micro-slots per AMW in a 6-well format[19]. For each AMW, $5 \times 10^6$ undifferentiated human iPSCs were seeded in 3 mL CCM+ supplemented with 100 ng/ml bFGF[84] and 10 μM ROCK inhibitor Y-27632 (Selleckchem, #S1049). iPSCs formed uniform embryoid bodies (EBs) during the initial 24 h on AMW and were harvested and transferred to suspension culture in 15-cm dishes and placed on an orbital shaker at 60 rpm. Suspension EBs were cultivated for further 3 days in CCM+ supplemented with 100 ng/ml bFGF before the start of cardiac differentiation. Differentiation was induced with the exchange of medium to RPMI 1640 supplemented with 1× B-27 supplement without insulin (RB-, Thermo Fisher Scientific, #A18956-01). The GSK-3 inhibitor CHIR99021 (Selleckchem, #S1263) was added at 4–6 μM (depending on the iPSC line) for the first 24 h of differentiation, thereafter differentiations were kept in RB- without small molecule inhibitors until day 3 (d3) before the WNT inhibitor IWR-1 (4 μM, Sigma, #I0161) was added for 48 h. The first contracting EBs were observed between d5 and d7. From d7–d10, EBs were cultivated with RPMI 1640 supplemented with 1× B-27 supplement with insulin (RB+, Thermo Fisher Scientific, #17504-044). Afterward, a metabolic selection was performed for 10d (until d20) to eliminate all non-CM in RPMI minus glucose (Thermo Fisher Scientific, #11879-020) supplemented with human albumin (Sigma, #A0237), sodium DL-lactate (Sigma, #L4263), and L-ascorbic acid-2-phosphate (Sigma, #A8960)[87]. From d20 to d40, CBs were kept in RB+ medium. Depending on the experimental design, inhibition of MEK was started on d12 of differentiation by supplementing the medium with 0.1 and 0.2 μM PD0325901 (Sigma, #PZ0162). Flow cytometric assessment of CM content was done on d20 by staining against cardiac troponin T (Life Technologies) after dissociation of aggregates with Stemdiff CM dissociation kit (STEMCELL Technologies) after adding Accutase to the enzyme mix (1:2). Aggregates with a purity of >95% were termed CBs and further cultivated.

**Single-cell suspensions of CBs and Flow cytometry.** For flow cytometric analysis, single-cell suspensions of undifferentiated iPSCs were obtained with Accutase, and cells were washed with ice-cold phosphate-buffered saline (PBS)$^{-/-}$. CBs were dissociated into single cells by incubation with Versene (EDTA-Solution, Thermo Fisher Scientific, #15040066) for 10 min in a Thermomixer at 37 ℃. Thereafter, TrypLE (Thermo Fisher Scientific, #A1285901) was added and samples were incubated for an additional 10 min at 37 ℃ and 1200 rpm until the cellular aggregates have dissolved. Both cell types were fixed in 4% formaldehyde (Carl Roth, #P087.1) for 10 min on ice and permeabilized with 90% ice-cold methanol for 20 min followed by a blocking step with 1.5% bovine serum albumin and 2.5% goat or donkey serum diluted in PBS for 1 h, at 4 ℃. Cells were stained with primary antibodies.

**BCT preparation, culture, and measurements.** On day 21, non-MEKi-treated CBs were dissociated as described above and a previously described protocol was used to generate BCTs[88]. In brief, a mixture of rat tail collagen type I (Cultrex) and Matrigel™ (Life Technologies) was mixed with CMs and gamma-irradiated human foreskin fibroblasts (HFFs) and poured into silicone molds (5 × 5 × 10 mm, w/d/h) with two horizontal titanium rods that serve as suspensions for the developing tissue at a distance of 6 mm, resulting in BCTs with a volume of 250 μL each containing $10^6$ CMs, $10^5$ HFFs, collagen type I (230 μg), and 10% Matrigel (Fig. S6a). A culture medium composed of DMEM (25 mM glucose), 10 % horse serum, 2 mM L-glutamine (all Life Technologies), 10 μg/mL insulin, and 100 U/mL penicillin and 100 μg/mL streptomycin (Sigma-Aldrich) was exchanged every other day, and the medium for RAF1[S257L] BCTs was additionally supplemented with 0.1 μM PD0325901 (Fig. S6b). Starting from day 10 of tissue culture, BCTs were stretched by 200 μm increments every fourth day to support tissue maturation, resulting in a final tissue length of 6.8 mm (Fig. S6c). On days 27 and 28, CSAs were determined by micrographs of the rod-shaped BCTs (Fig. S7a), and

physiological measurements were performed in a custom-made bioreactor system in a BCT culture medium as follows: Tissue samples were placed in a culture vessel allowing for simultaneous recording of spontaneous and electrically paced tissue contractions to determine spontaneous beating frequencies and contraction forces, respectively (Fig. S6d, e). For the latter, electrical field stimulations of 25 V biphasic pulses (±5 ms) were applied five times at a frequency slightly higher than the spontaneous beating frequency but not below 1 Hz. To measure maximum contraction forces, tissue samples were stretched by 200 µm increments until a total additional preload of 1.2 mm was reached (Fig. S7b). CSAs were used to determine individual maximum contractile tensions (mN/mm$^2$). The shape of contraction and contraction kinetics, i.e., time to contraction peak (TTP) and time to 80% relaxation (TTR) were assessed at the preload step, where the maximum contraction force was recorded for each sample, i.e., $L_{max}$ with 1 Hz of electrical pacing. Inotropic responses to increasing calcium concentrations and energy efficiency were determined after complete calcium removal to eliminate spontaneous contractions. For this, tissue samples were washed with Tyrode's solution without calcium (120 mM NaCl, 5.4 mM KCl, 1 mM MgCl$_2$, 0.4 mM NaH$_2$PO$_4$, 22 mM NaHCO$_3$, 5 mM Glucose). Calcium concentration was increased stepwise in Tyrode's solution (0.1, 0.2, 0.4 0.6, 0.8, 1.0, 1.5, 2, 2.5, 3.0, 4.0, 5.0 mM) and the tissue samples were paced electrically as described above, however, the BCTs were measured at their original length of 6.8 mm. Contraction forces were normalized to values that each tissue sample reached at this length in a normal culture medium containing 25 mM glucose, measured just prior to washing with Tyrode's solution.

**Karyotype analysis**. After treatment of undifferentiated iPSCs with a final concentration of 0.1 µg/mL Colcemid (Thermo Fisher Scientific) for 2 h, cells were detached with trypsin/EDTA (0.05/0.02%, Biochrom). After centrifugation, the pellet was resuspended in hypotonic solution (0.32% KCl with 0.2% (v/v) fetal calf serum) and incubated for 15 min at 37 °C. Cells were fixed in ice-cold methanol/acetic acid (3:1). G-banding was performed according to Seabright protocol[89]. Karyograms were imaged using the IKAROS software of MetaSystems (Altlußheim, Germany). The chromosome arrangement was investigated as previously described[90].

**Quantitative real-time reverse transcriptase polymerase chain reaction**. Cells were lysed using TRIzol™ (Ambion, Thermo Fisher Scientific, Germany), and total RNA was extracted *via* phenol-chloroform extraction. The remaining genomic DNA contaminations were removed using the DNA-free™ DNA Removal Kit (Ambion, Thermo Fisher Scientific, Germany). DNase-treated RNA was transcribed into complementary DNA (cDNA) using the ImProm-II™ reverse transcription system and oligo-dT as a primer (Promega, Germany). Quantitative real-time reverse transcriptase polymerase chain reaction (qPCR) was performed using SYBR Green (Thermo Fisher Scientific, #4309155 Germany). Primer sequences are listed in Table S1. The $2^{-\Delta\Delta Ct}$ method was employed for estimating the relative mRNA expression levels and $2^{-\Delta Ct}$ for mRNA levels. Among six different housekeeping genes that we tested, *HPRT1* showed the lowest variation among different cell lines and conditions. Therefore, *HPRT1* was used for normalization in our qPCR analysis.

**Immunoprecipitation and immunoblotting**. CBs and BCTs were lysed in 50 mM Tris-HCl pH 7.5; 100 mM NaCl, 2 mM MgCl$_2$, 0.5% NP-40, 10% glycerol, 20 mM beta-glycerol phosphate, 1 mM orthoNa$_3$VO$_4$, and EDTA-free protease inhibitor (Roche, Germany, #11873580001). Immunoprecipitation was carried out using the same buffer without NP-40, specific antibodies, and a non-specific anti-IgG for the negative control, respectively. A mixture of the lysates, the respective antibodies, and beads was rotated for 1 h at 4 °C. Beads were washed five times and heated in SDS/Laemmli sample buffer at 95 °C for 10 min before being subjected to immunoblotting[91].

**IHC and immunocytochemistry**. From formalin-fixed tissue, 3 µm sections were stained with H&E and Masson trichrome (Trichrome II Blue staining kit at Nexus special stainer; Roche). Immunohistochemical analysis was performed on cryosections and paraffin sections using a Bench Mark XT automatic staining platform (Ventana, Heidelberg, Germany) with the primary antibodies[92]. After formalin-fixation of the surgically removed tissue, 3 µm sections were stained with hematoxylin and eosin. Immunohistochemical analysis was performed on cryosections and paraffin sections using a Bench Mark XT automatic staining platform (Ventana, Heidelberg, Germany), using the primary antibodies listed in Table S2. The sections were examined using a Nikon Eclipse 80i equipped with a DS-Fi1 camera. IHC of BCT samples was carried out using BCTs embedded and snap-frozen in Tissue-Tek® OCT resin (Sakura Finetek, #4583). The cryo-blocks were sliced into 8 µM thick sections. Sections were fixed in 4% paraformaldehyde (Carl Roth, #P087.1) in 0.1 mol/L sodium phosphate buffer pH 7.4 for 10 min. Washing was performed in PBS and 0.2% saponin/PBS. After blocking with 10% normal goat serum (NGS) in 0.2% saponin/PBS for 1 h, primary antibodies were incubated overnight at 4 °C. Secondary antibodies were incubated for 3 h at room temperature in the dark. The sections were mounted with ProLong® Gold Antifade Mountant, containing ProLDAPI (#P-36935, Invitrogen). Slides were analyzed with a ZEISS Airyscan LSM 880 confocal microscope (Center for Advanced Imaging,

Heinrich Heine University, Düsseldorf, Germany). Pictures were taken with 63× objective and analyzed using ZEN 3.2 (blue edition) by Carl Zeiss AG.

**Analysis of cardiomyocyte surface area**. The CM surface was analyzed using a Java-based open-source image processing software ImageJ. Software pixel length was calibrated to the µm scale bar on the used confocal images. After background elimination, the region of interest tool (ROI) was used to mark the borders of single cells. For each condition, multiple confocal images were evaluated to evaluate a sufficient number of images. Finally, the cell area was extracted and compared between the conditions.

**Analysis of the sarcomere organization**. The Sota software was used to quantify sarcomere organization. We selected the folder with the images to analyze and input the resolution (pixels per micron). The background subtraction option was selected. Segmentations 2 × 2y were chosen. Other settings were as follows: offset distance (6 µm), rolling ball size (2.5), disregard lowest GLCM-bin (3), and maximum sarcomere length (3 µm). We performed an analysis and checked the output graphs to see whether they looked like the ones we expected from software guidelines.

**Transmission electron microscopy**. Small primary cardiac tissue samples were fixed with 6% glutaraldehyde/0.4 M PBS and processed with a Leica EM TP tissue processor. CBs were fixed with 3% glutaraldehyde/0.1 M Cacodylate buffer. The cell pellets were processed by hand according to the automated tissue processor. For electron microscopy of the small cardiac tissue samples and CBs, ultrathin sections were contrasted with 3% lead citrate trihydrate with a Leica EM AC20 (Ultrastain kit II) and were examined using a ZEISS EM 109 transmission electron microscope equipped with a Slowscan-2K-CCD-digital camera (2K-wide-angle Sharp: eye), while CBs were imaged using a Hitachi H-7100.

**Measurement of calcium cycling**. iPSC-derived CBs were dissociated and grown on gelatin-coated coverslips for up to 7 days before loading with the fluorescent Ca$^{2+}$ indicator Fura-2 by adding 1 µg Fura-2-AM/mL cell medium. After a 15 min incubation at 37 °C, cells were washed in a pre-warmed medium (37 °C). A dual excitation (340 nm and 380 nm) fluorescence imaging recording system was used to measure Ca$^{2+}$ transients of paced (0.5 Hz) and spontaneously beating cells (HyperSwitch Myocyte System, IonOptix Corp., Milton, MA, USA). Data were acquired as the ratio of measurements at 340 and 380 nm and analyzed using IonWizard software (Version 6.4, Ion Optix Corp).

**Statistics and reproducibility**. Statistical analysis was performed using GraphPad Prism software (version 9 for MacOS and Windows; GraphPad Software). Unless otherwise stated, values are given as mean and standard deviation. The statistical significance of differences between the two groups was determined by an unpaired, two-tailed Student *t*-test and, if there were more than two groups, by a one-way analysis of variance with Bonferroni's multiple comparison test. Differences were considered statistically significant at a value of $P < 0.05$ and marked with *.

**Reporting summary**. Further information on research design is available in the Nature Portfolio Reporting Summary linked to this article.

## Data availability

Source data for figures can be found in Supplementary Data 1 and all other data are available upon reasonable request.

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

## Acknowledgements

We are grateful to Dr. Ehsan Amin and Prof. Dr. Thomas Wieland for helpful advice, and stimulating discussions. We are grateful to the CFEM (Core Facility Electron Microscopy) of the medical faculty of Heinrich Heine University Düsseldorf. We gratefully thank the entire team from the Stem Cell Unit, University Medical Center Göttingen, for excellent technical assistance in iPSC generation and characterization. The authors thank Louisa Habich, Sarah Nourmohammadi, Hannah Schlierbach, David Skvorc, Marion Möckel, and Kerstin Leib for excellent technical assistance in iPSC differentiation, CM selection, and BCT production. We are grateful for the support in bioreactor programming and operation from Dr. Jan Patrick Pietras. We thank and give credit to "BioRender.com" which was used for the design and creation of Fig. 1a. This study was supported by the German Federal Ministry of Education and Research (BMBF)—German Network of RASopathy Research (GeNeRARe, grant numbers: 01GM1902A, 01GM1902C, 01GM1902D, 01GM1902F); the German Center for Cardiovascular Research (DZHK); the European Network on Noonan Syndrome and Related Disorders (EJP-RD; NSEuroNet, grant numbers: 01GM1921A, 01GM1921B, 01GM1807); Deutsche Gesellschaft für Muskelkranke (DGM) e.V. (Sc22/11); German Research Foundation (DFG): grant numbers AH 92/8-1 to MRA, Ci216/2-1 to ICC, SFB 974, P3 to MRA and B9 to ASR, SFB1002 S01 to LC, SFB1116-1/2 TPA02 to MK and JS, GRK 2578 to ASR and IRTG 1902 P6 to MRA.

## Author contributions

Conception, design, and writing: M.R.A., G.K., S.N.R., F.H., and F.B. Development of methodology: S.N.R., F.H., F.B., M.B., J.D., K.K., M.V., M.K., A.K.B., A.B., D.S., R.A., J.P.S., G.K., L.C. Acquisition of data, analysis, and interpretation of data: S.N.R., F.H., F.B., A.V.B., M.B., J.D., F.F., S.K., A.K.B., A.B., D.S., A.V.K., A.S.R., J.P.S., A.S., B.G., I.K., M.T., G.K. Administrative, technical, or material support: M.R.A., G.K., M.Z., M.K., A.G., M.J.W., A.S., A.S.R., A.H., J.S., R.P.P., I.C.C., J.P.S., L.C. Study supervision and coordination: M.R.A., G.K., and M.Z.

## Funding

## Competing interests

B.D.G.'s institution receives royalties from genetic testing for RAF1 for Noonan syndrome from Correlegan, LabCorp, GeneDx, and Prevention Genetics. B.D.G. is a consultant for Day One Biopharmaceuticals and was recently a consultant for BioMarin. He received a sponsored research award from Onconova. M.Z. is a member of a scientific advisory board for Day One Biopharmaceuticals.

## Ethical approval and consent to participate

Blood samples and dermal fibroblasts were obtained by skin biopsy from two unrelated female patients with Noonan syndrome carrying the heterozygous substitution c.770 C>T in exon 7 of RAF1 under protocols concerning research with biomaterials approved by the institution's ethics committees (Justus-Liebig-University Giessen, Germany: AZ258/16; Otto von Guericke University Medical Center Magdeburg: 173/14; University Medical Center Göttingen: 10/9/15). Research with biomaterials was approved by the ethical committee of the Justus Liebig University of Giessen. Written informed consent was given by both parents of both individuals. One human iPSC line that was used in this study as a control, was generated from the human foreskin fibroblast-1 (HFF-1) cell line (ATCC).

## Additional information

[1]Institute of Biochemistry and Molecular Biology II, Medical Faculty and University Hospital Düsseldorf, Heinrich Heine University Düsseldorf, Düsseldorf, Germany. [2]Stem Cell Biology and Regenerative Medicine Research Group, Institute of Biotechnology, Ferdowsi University of Mashhad, Mashhad, Iran. [3]Clinic for Cardiothoracic and Vascular Surgery, University Medical Center Göttingen, Göttingen, Germany. [4]German Center for Cardiovascular Research (DZHK), partner site Göttingen, Göttingen, Germany. [5]Institute of Human Genetics, University Hospital, Otto von Guericke-University, Magdeburg, Germany. [6]Stem Cell Unit, Clinic for Cardiology and Pneumology, University Medical Center Göttingen, Göttingen, Germany. [7]Cluster of Excellence "Multiscale Bioimaging: from Molecular Machines to Networks of Excitable Cells", University of Göttingen, Göttingen, Germany. [8]Institute of Neuropathology, Justus Liebig University Giessen, Giessen, Germany. [9]Institute of Biochemistry and Molecular Biology I, Medical Faculty and University Hospital Düsseldorf, Heinrich Heine University Düsseldorf, Düsseldorf, Germany. [10]Department of Child Neurology, Justus Liebig University Giessen, 35392 Giessen, Germany. [11]Institute of Cardiovascular Physiology, Medical Faculty and University Hospital Düsseldorf, Heinrich Heine University Düsseldorf, Düsseldorf, Germany. [12]Institute of Pharmacology, Medical Faculty and University Hospital Düsseldorf, Heinrich Heine University Düsseldorf, Düsseldorf, Germany. [13]Department of Medicine and Robert M. Berne Cardiovascular Research Center, University of Virginia, Charlottesville, VA 22908, USA. [14]Institute of Comparative Molecular Endocrinology, University of Ulm, Helmholtzstrasse 8/1, 89081 Ulm, Germany. [15]Mindich Child Health and Development Institute and Departments of Pediatrics and Genetics and Genomic Sciences, Icahn School of Medicine at Mount Sinai, New York, NY 10029, USA. [16]Molecular Genetics and Functional Genomics, Ospedale Pediatrico Bambino Gesù, IRCCS, 00146 Rome, Italy. [17]These authors contributed equally: Saeideh Nakhaei-Rad, Fereshteh Haghighi, Farhad Bazgir. ✉email: martin.zenker@med.ovgu.de; george.kensah@med.uni-goettingen.de; reza.ahmadian@uni-duesseldorf.de

