## [Peer Review File · Communications Biology]

Reviewers' comments:

Reviewer #1 (Remarks to the Author):

In this study, Nakhaei-Rad et. al., engineered two different three-dimensional (3D) human cardiac cell models, cardiac bodies (CBs) and bioartificial cardiac tissues (BCTs) to elucidate cardiac-specific impacts of RAF1S257L. The manuscript is very well-written and easy to follow. However, a similar study has been published in *Circulation* (Jaffre et al., 2019) using RAF1S257L/+ and CRISPR corrected isogenic control iPSC derived cardiomyocytes to model NS-associated-HCM and revealed that hyper-activation of MEK1/2, but not ERK1/2, caused myofibrillar disarray, whereas the enlarged cardiomyocyte phenotype was caused by increased ERK5 signaling. Additionally, this paper also reported dysregulation of gene expression of MYH7. Therefore the true novelty of this study needs to be addressed.

Major points:

- 1) Certain signaling events were only tested in one RAF1S257L cell line lacking statistical significance from biological repeats. It was also unclear how many RAF1S257L cell lines were used in most of the figures.
- 2) In Figure 3, the sarcomere alignment needs to be quantified in multiple cells/fields. The alignment of the mutant line seems much better in panel F than panel A.
- 3) The authors did not provide direct evidence showing MYH7 and NPPB are YAP targets in cardiomyocytes.

Minor points:

- 1) Some technical details could be shortened in the Results section and described in the Methods (e.g., To this goal, d40 CBs were dissociated, and single CMs were seeded on coverslips. Thereafter, CMs were fixed and...).
- 2) Figure legend of Figure S1, second line from the bottom, FOXA2 is misspelled as "FOXOA2".
- 3) The data regarding "iPSC-derived CMs had the ability to increase in size in response to hypertrophic stimuli" should be in main data.

Reviewer #2 (Remarks to the Author):

The authors use a combination of iPSC-derived cardiospheres and engineered heart tissue to study how Noonan-syndrome associated disruption of normal RAF1 signaling leads to cardiomyopathy phenotypes in vitro. While Noonan syndrome has been associated with HCM in patients, the phenotypes observed here – many of which lead to less calcium intake and lower contractility – appear more closely tied to previous in vitro models of Dilated, rather than Hypertrophic Cardiomyopathy. This may reflect that the process through which signaling-associated mutations induce hypertrophic phenotypes is distinct from the way sarcomere mutations more commonly tied to HCM. The work is overall of high quality and of general interest to the field.

Given potential impact of the conclusions of this work, I do have some concerns related to how much of the observed phenotypes truly emerge from RAF1 signaling perturbations, versus how much they relate to line-to-line variation in iPSC. I also have concerns about the strength of conclusions related to contractility studies (Figure 6), as it is unclear whether peak forces are being compared in tissues beating at the same, or different frequencies. Besides these overreaching concerns, I have a few additional experimental concerns. Overall I find the work exciting but view it as requiring some more experimental evidence to be conclusive.

Specific experimental critiques:

My first two questions address the possibility that phenotype changes seen here are simply due to

line-to-line variability (ie, they could also be observed with another "wild type" line and are not due to RAF1). While the MEK inhibition studies partially address this possibility, additional experiments are needed:

1) While I appreciate the vast amount of work that went into the physiology studies, at least some subset of these (calcium handling, tissue contractility) must be repeated in the isogenic control (RAF1iso, currently described in Fig. 2F). This would go a long way toward providing conclusive evidence that the differences between WT and RAF1 mutant lines are truly reflective of RAF1 function and not line-to-line variability.

2) While the studies here were only done with lactate purified cardiomyocytes: before lactate, based on flow cytometry for cardiac troponin on dissociated cardiomyocytes, is the differentiation efficiency similar amongst the lines tested here? Are there cardiomyocyte markers that are very similar across all the iPSC-cardiomyocytes regardless of genotype? Or is there reason to believe there is a differentiation defect that may be unrelated to RAF1?

Other experimental points:

3) A major point made early on is "A remarkable and unprecedented finding in the present study is the atypical I-bands, in both RAF1S257L CMs as well as the heart biopsy samples from the corresponding individual with the heterozygous RAF1S257L variant." Data should be provided to back up this statement, or a reference to a previous study showing that primary heart biopsy data.

4) It is unfair to compare the contractility of tissues beating at different rates (Fig. 6B). I strongly recommend these either be repeated with electrical field pacing. If this is not possible, then at least the authors can report contractility only for the subset of tissues that have a very similar spontaneous beating frequency (a significant number of tissues of each genotype and +/- MEK inhibitor appear to have a frequency near 0.75 Hz).

5) Differences in contractility kinetics (rise and relaxation times) should be assessed and reported – as in comment above, this should only be done for tissues beating at the same or very similar rate, but ideally in paced tissues.

Minor experimental critiques:

1) For all physiology studies (calcium handling and contractility): please provide specific examples of (e.g. representative calcium and force traces).

2) Please show a representative BCT for each genotype in Figure 5, in part to clarify that these measurements reflect BCT sections and not cardiomyocytes plated into 2D

3) In panel 5G, two of the data points in the WT set seem to dominate the response. Have the authors performed an outlier analysis on these data?

Writeup and discussion

1) Many of the features of the disease observed here, including lower calcium intake and hypocontractility, combined with larger cell size, are more similar to literature reports of dilated, rather than hypertrophic cardiomyopathy. This is an important difference that should be highlighted and discussed in the context of other in vitro cardiomyopathy models (for example, of dilated cardiomyopathy, sarcomere-linked hypertrophic cardiomyopathy), especially those that have used iPSC.

2) Any disadvantages of mouse models of Noonan syndrome should be briefly provided as a rationale for using this system that produces human, albeit immature cardiomyocytes.

Responses to the reviewers' comments

Reviewer #1 (Remarks to the Author):

In this study, Nakhaei-Rad et. al., engineered two different three-dimensional (3D) human cardiac cell models, cardiac bodies (CBs), and bioartificial cardiac tissues (BCTs) to elucidate cardiac-specific impacts of RAF1^{S257L}. The manuscript is very well-written and easy to follow.

Response: Thank you very much for reviewing our manuscript. My colleagues and I appreciate your positive and constructive comments, which are very valuable and important as they significantly increase the quality and clarity of our manuscript. We have responded to all comments point-by-point (see below). The most important changes are marked in yellow in the revised version of the manuscript.

However, a similar study has been published in *Circulation* (Jaffre et al., 2019) using RAF1^{S257L/+} and CRISPR-corrected isogenic control iPSC-derived cardiomyocytes to model NS-associated-HCM and revealed that hyper-activation of MEK1/2, but not ERK1/2, caused myofibrillar disarray, whereas the enlarged cardiomyocyte phenotype was caused by increased ERK5 signaling. Additionally, this paper also reported dysregulation of gene expression of *MYH7*. Therefore, the true novelty of this study needs to be addressed.

Response: Thank you for raising this important point. We are aware of the study by Jaffré et al. and the similarity of some of our findings, also reported by this group, speaks to the validity of the main findings of both studies. However, our study has raised several novel aspects as follows:

1) More informative 3D cardiac models to overcome limitations of other models, including

2D: We applied two different 3D cardiac models, cardiac bodies (CBs) and bioartificial cardiac tissues (BCTs), around 50 days post differentiation. We abandoned 2D cardiomyocytes at an early stage of our study because the cell size was heterogeneous and the morphology was rather flat than that of aligned-cardiac cells and proceeded with the 3D cell models (*Figure R1*). The advantage of these 3D models is their higher homogeneity and more mature properties due to cell-cell communications, constant isometric strain, and the presence of scaffold proteins (Goldfracht et al., 2019; Dahlmann et al., 2013; Kensah et al., 2013).

2) First report of an abnormality in the I-band region in Noonan syndrome associated with an isoform shift of titin from a longer (N2BA) to a shorter isoform (N2B):

For the first time, we were able to look in more detail at the ultrastructure of RAF1^{S257L} iPSC-derived CMs (Figures 3 and 4), leading to the unprecedented discovery of the I-band shortening. We found an isoform shift in titin from a longer (N2BA) to a shorter isoform (N2B), potentially explaining the shortening of the I-band, which can have a crucial influence on the flexibility and stiffness of the sarcomeres along the Z-line.

3) Evidence for the use of an iPSC-derived 3D cardiac model brings us one step closer to recapitulating the organ state:

As proof of the quality of our 3D cardiac model systems, we found disorganized sarcomere structures with shortened I-bands together with a thickened Z-line also in the heart biopsy of one of the patients from whom the biopsy sample was available. This observation shows that our system resembles the primary myocardium (heart tissue) in this respect and is thus one step closer to recapitulating the organotypic features *in vitro*.

4) Quantitative functional studies of the 3D cardiac model systems:

In addition, the generation of BCTs allows us to apply defined preload to the myocardium enabling quantitative studies on tissue samples. The cardiac cells in BCTs, in contrast to the 2D system

(Figure R1), are aligned and unidirectionally contract simultaneously, allowing us to determine the sum of the contraction force of each cardiac cell.

Figure R1. The comparison of the two different differentiation models, 2D vs. 3D (Bioartificial cardiac tissues). Scale bar: 50 μ m.

5) Providing complementary molecular details to explain the observed cardiac abnormalities (SERCA2a to PLN ratio, BNP level, p-38, and Hippo-YAP): Our study provided the following additional molecular details: (a) We observed a reduced calcium transition and found that the SERCA2a-to-PLN ratio is reduced in the $RAF1^{S257L}$ -CMs. (b) Cellular expression and secretion of BNP were significantly increased in $RAF1^{S257L}$ cardiomyocytes, which we found to be regulated by the MAPK signaling pathway. Overall, our work adds a broader spectrum of molecular and physiological explanations for $RAF1^{S257L}$ -induced HCM, and the main similarity with the previous study is that both studies investigated the same frequently occurring mutation, but as described above, our system, methods, cardiac model, and results differ.

7) Cardiac-specific $RAF1$ signaling to ERK5 and calcineurin-NFAT in HCM: HCM has been reported to correlate with ERK5 activation (reviewed in (Gallo *et al.*, 2019): (i) Cardiomyocyte-specific ERK5 KO mice showed a lower hypertrophic response and induction of apoptosis during pressure overload; (ii) stimulation of ERK5 was associated with eccentric and detrimental hypertrophy in an animal model of long-term intermittent cardiac hypoxia; (iii) ERK5 activation is promoted by AngII hypertrophic detrimental stimulus. Jaffré *et al.* argue for cardiac-specific $RAF1$ signaling to ERK5 and calcineurin-NFAT in HCM although they did not observe differences in ERK5 phosphorylation. They instead inhibited the MEK5/ERK5 pathway with BIX02189, an inhibitor of both MEK5 (IC50 = 1.5 nmol/L) and ERK5 (IC50 = 59 nmol/L), and showed that treatment of $RAF1^{S257L}$ CMs with BIX02189 significantly reduced cell surface area (Jaffré *et al.* 2019).

We did not pursue this issue further for the following reasons: (i) We also analyzed ERK5 phosphorylation and observed hardly any pattern of p-ERK5 with $RAF1^{S257L}$ with and without MEKi as well as $RAF1^{corr}$ (Fig. R2). (ii) ERK5 activation could be EGFR-dependent but RAS-independent, as well as independent of the RAF-MEK-ERK axis (Tatake *et al.*, 2008). (iii) The small molecule inhibitor BIX02189 used in Jaffré *et al.* also has an inhibitory effect on the TGFbeta receptor and SMAD pathway (Park *et al.*, 2016). Given this important point and the known importance of SMAD1/5 activity as a driver of pathogenic cardiac hypertrophy, it may be possible that the observation by Jaffré *et al.* with BIX02189 could be due to inhibition of the TGFbR/SMAD pathway.

Figure R2. Representative immunoblots of ERK5 vs. p-ERK5 using cell lysates from $RAF1^{S257L}$, $RAF1^{S257L}$ + MEKi, and $RAF1^{corr}$ CBs (d24), using a monoclonal rabbit anti-ERK5 antibody (#12950S, Cell Signaling) and a polyclonal rabbit anti-p- $ERK5^{Thr218/Tyr220}$ (#3371S, Cell Signaling). A monoclonal mouse anti-GAPDH antibody (#398600, Thermo Fisher Scientific) was used as a loading control.

As requested, we have included the following texts in the Introduction and Discussion sections of the revised manuscript

Introduction section (pages 2-3):

In this study, we generated and used two different three-dimensional (3D), human cardiac cell models, cardiac bodies (CBs), and bioartificial cardiac tissues (BCTs) in the presence and absence of a MEK inhibitor (PD0325901) to elucidate the cardiac-specific impacts of RAF1^{S257L} on sarcomere structure, contractile behavior, Ca²⁺ handling, and intracellular signaling. To this end, two iPSC lines derived from two different individuals carrying the recurrent *RAF1* c.770C>T (p.Ser257Leu) variant and three independent control iPSC lines were used (Fig. 1A). Our study provided the following important new findings: (i) Monogenic RAF1^{S257L}-related ultrastructural abnormalities at the sarcomere, including shortening of the I-band, are associated with an isoform shift of titin from a longer (N2BA) to a shorter isoform (N2B). This phenotype is associated with disrupted sarcomere structures, which were also found in a heart muscle biopsy of one NS patient. (ii) Elevated BNP levels, which is a common indicator of HCM, were observed in our model. (iii) SERCA2a and L-type calcium channel, two main regulators of intracellular calcium transients, were downregulated, resulting in a shift in the SERCA2/PLN ratio. Calcium transient amplitudes in RAF1^{S257L}-CMs (cardiomyocytes) were also smaller consistent with reduced amounts of free calcium that cycle between the sarcoplasmic reticulum and the cytosol. These results, therefore, suggest a role in the aberrant contraction frequencies and contractile tensions in our 3D model. (iv) Functional analysis of the RAF1^{S257L}-BCTs revealed increased spontaneous contraction frequencies, myocardial thickening, lower contractile tensions, aberrant contraction kinetics, and a significant decrease in the energy efficiency of the working myocardium. (v) Increased MAPK, p38, and increased YAP signaling events in CBs contribute to altered functional behavior and calcium transients. (vi) Treatment with a MEK inhibitor (PD0325901) rescued most of the observed hypertrophic phenotype caused by RAF1 gain-of-function, which was validated using a gene-corrected isogenic control. Importantly, the critical structural findings of the *in vitro* models were consistent with the results of a myocardial biopsy of one of the NS individuals.

Discussion section (Page 12):

HCM has been reported to correlate with ERK5 activation (reviewed in (Gallo *et al.*, 2019)): (i) CM-specific ERK5 KO mice showed a lower hypertrophic response and induction of apoptosis during pressure overload; (ii) stimulation of ERK5 was associated with eccentric and detrimental hypertrophy in an animal model of long-term intermittent cardiac hypoxia; (iii) ERK5 activation is promoted by angiotensin II hypertrophic detrimental stimulus. Jaffré *et al.* argue for cardiac-specific RAF1 signaling to ERK5 and calcineurin-NFAT in HCM although they did not observe differences in ERK5 phosphorylation. They instead inhibited the MEK5/ERK5 pathway with BIX02189, an inhibitor of both MEK5 (IC₅₀ = 1.5 nmol/L) and ERK5 (IC₅₀ = 59 nmol/L) showed that treatment of RAF1^{S257L} CMs with BIX02189 significantly reduced cell surface area (Jaffré *et al.* 2019).

We did not pursue this further for the following reasons: (i) We also analyzed ERK5 phosphorylation and observed hardly any pattern of p-ERK5 with RAF1^{S257L} with and without MEKi as well as RAF1^{corr} (Fig. R2). (ii) ERK5 activation could be EGFR-dependent but RAS-independent, as well as independent of the RAF-MEK-ERK axis (Tatake *et al.*, 2008). (iii) The small molecule inhibitor BIX02189 also has an inhibitory effect on the TGFβ receptor (TGFβR) and SMAD pathway (Park *et al.*, 2016). Given this important point and the known importance of SMAD1/5 activity as a driver of pathogenic cardiac hypertrophy, it may be possible that the observation by Jaffré *et al.* with BIX02189 could be due to the inhibition of the TGFβR/SMAD pathway.

Major points:

1) Certain signaling events were only tested in one RAF1^{S257L} cell line lacking statistical significance from biological repeats. It was also unclear how many RAF1^{S257L} cell lines were used in most of the figures.

Response: For signal profiling, we used two RAF1^{S257L} cell lines from unrelated individuals with Noonan syndrome. In Figures 2D and E, we used the 7B10 (line 1). In Figure 2D, we used two biological replicates, of which only one blot was shown for each protein, and the statistical significance comes from two sets of cardiac differentiation experiments (biological repeats). In Figure S5, we also repeated the signaling experiments using a second isogenic cell line (isRASb1.17) from the iPSCs of the second RAF1^{S257L} individual with two biological replicates. The statistical details are given in the legend of the respective Figures. “n≤2” was a typo in the figure legends, which should actually read n≥2, representing at least two biological repeats. This has now been corrected in the figure legends.

2) In Figure 3, the sarcomere alignment needs to be quantified in multiple cells/fields. The alignment of the mutant line seems much better in panel F than in panel A.

Response: Thank you for pointing this out. The CMs in former Figure 3 (new Figure 4) were cultured for 47 days (Figure 4A) or 90 days (Figure 4F) post-differentiation to observe differences in their maturation state. Accordingly, we observed differences in the improved organization of the sarcomere structure of the RAF1^{S257L}-cardiac cells, which could be due to the maturation process that occurred during differentiation. In this context, in Figure 4A we have taken the images, 7 days after the dissociation of the cardiac bodies (CBs), while in Figure 4F the cells are almost two months further in differentiation (the information is indicated in the Figure legends). At your request, we have quantified the sarcomere structures of the more mature cells in Figure 4F. For this purpose, we used the "Sota software" developed for the quantification of sarcomeres (Stein et al., 2022). As shown in *Figure R3*, the organization values range from 0-0.3, where 0 stands for unorganized and 0.3 for organized sarcomeres. In the same line of evidence, quantification data here confirm that mutant CMs exhibited less organized and aligned sarcomere structures. We have added these data as a new Figure 4G in the revised manuscript.

Figure R3. The quantifications of the sarcomeres in Figure 4F for cTNT. Indicated the sarcomere organization (left panel) and alignment (right panel) in wild-type and RAF1^{S257L} cardiomyocytes (CM), which were obtained by dissociating the cardiac bodies (CBs).

We included the following text in the Materials and Methods section “Analysis of cardiomyocyte surface area” (page 7):

The Sota software was used to quantify sarcomere organization. We selected the folder with the images to analyze and input the resolution (pixels per micron). The background subtraction option was selected. Segmentations 2x2y were chosen. Other settings were as follows: offset distance (6 μm), rolling ball size (2.5), disregard lowest GLCM-bin (3), and maximum sarcomere length (3 μm). We performed an analysis and checked the output graphs to see whether they looked like the ones we expected from software guidelines.

3) The authors did not provide direct evidence showing MYH7 and NPPB are YAP targets in cardiomyocytes.

Response: Thank you for pointing out this issue. Two different groups have reported that the transcriptional regulation of *MYH7* and *NPPB* genes appears to be under the control of YAP-TEAD1 (Stein *et al.*, 2015; Akerberg *et al.*, 2019). Akerberg *et al.* analyzed the chromatin occupancy by different transcription factors GATA4, NKX2-5, MEF2A, MEF2C, SRF, TBX5, and TEAD1, using CHIP-Seq data. They identified TEAD1-binding peaks for *MYH7* and *NPPB* genes (Akerberg *et al.*, 2019). These studies are now included in the Discussion section on page 12 where the sentence “We also observed increased expression of YAP targets, such as *MYH7* and *NPPB* in *RAF1*^{S257L} CBs.” is now rephrased: “Two different groups have reported that the transcription of *MYH7* and *NPPB* genes appears to be under the control of YAP-TEAD1 (Stein *et al.*, 2015; Akerberg *et al.*, 2019). Akerberg *et al.* analyzed the chromatin occupancy by different transcription factors GATA4, NKX2-5, MEF2A, MEF2C, SRF, TBX5, and TEAD1, using CHIP-Seq data. They identified TEAD1-binding peaks for *MYH7* and *NPPB* genes (Akerberg *et al.*, 2019). Similarly, we observed increased expression of *MYH7* and *NPPB* in *RAF1*^{S257L}-CBs, which correlates with our signaling data and may be due to increased active YAP levels in *RAF1*^{S257L}-CBs.”

Minor points:

1) Some technical details could be shortened in the Results section and described in the Methods (e.g., To this goal, d40 CBs were dissociated, and single CMs were seeded on coverslips. Thereafter, CMs were fixed and...).

Response: As indicated in the revised manuscript, we have deleted various technical details from the "Results" section and transferred them to the "Materials and Methods" section where necessary.

2) Figure legend of Figure S1, second line from the bottom, FOXA2 is misspelled as “FOXOA2”.

Response: Thank you for pointing out this error, which has now been corrected in the revised manuscript.

3) The data regarding "iPSC-derived CMs could increase in size in response to hypertrophic stimuli" should be in the main data.

Response: As suggested, we have inserted Figures S4C-F into the revised Figures 2B-E of the revised main manuscript.

Reviewer #2 (Remarks to the Author):

The authors use a combination of iPSC-derived cardiospheres and engineered heart tissue to study how Noonan-syndrome-associated disruption of normal *RAF1* signaling leads to cardiomyopathy phenotypes in vitro. While Noonan syndrome has been associated with HCM in patients, the phenotypes observed here – many of which lead to less calcium intake and lower contractility – appear more closely tied to previous in vitro models of Dilated, rather than Hypertrophic Cardiomyopathy. This may reflect that the process through which signaling-associated mutations induce hypertrophic phenotypes is distinct from the way sarcomere mutations are more commonly tied to HCM. The work is overall of high quality and general interest to the field.

Response: Thank you very much for reviewing our manuscript. My colleagues and I are pleased with the very positive response and appreciate your constructive comments, which are very valuable and important as they substantially improve the quality and clarity of our manuscript. We have responded to all comments point-by-point (see below). The most important changes are marked in yellow in the revised version of the manuscript.

Given the potential impact of the conclusions of this work, I do have some concerns related to how much of the observed phenotypes truly emerge from RAF1 signaling perturbations, versus how much they relate to line-to-line variation in iPSC.

Response: Thank you for bringing this issue to our attention. We generated iPSC^{S257L}-iPSC lines (7B10 and isRASb.17) from two unrelated individuals with Noonan syndrome in two different laboratories using non-integrating reprogramming protocols. We observed similar results in both iPSC^{S257L} lines (Figures 2-4 and S5), including signaling profiles, sarcomere abnormalities, I-band shortening, increased NPPB at the mRNA and protein levels, and disruption of MYH6/7 and SERCA/PLN ratios. The next question was whether these observations were from the RAF1-MEK-ERK axis or another RAF1 target. Therefore, we treated cell lines 1 and 2 with a MEK inhibitor (MEKi). For MEKi-treated cardiac cells as well as the isogenic gene-corrected control of isRASb.17, the observed phenotypes for I-band shortening, signaling profiles including p-ERK/p-p38/p-YAP, BNP/NPPB levels, and MYH6/7 and SERCA/PLN ratios were restored. All in all, our coherent results underline that there are no striking differences between the lines and the results are consistent between unrelated patient-derived iPSCs.

I also have concerns about the strength of conclusions related to contractility studies (Figure 6), as it is unclear whether peak forces are being compared in tissues beating at the same, or different frequencies.

Response: We are glad to have the opportunity to clarify this very important detail, as the Methods section did not mention how exactly these force measurements were performed. To support clarity, we have outlined the generation, maturation, and measurement procedures graphically in an additional supplementary Figure S6.

Figure S6. Overview of bioartificial cardiac tissue (BCT) generation, culture, treatment, and measurement. (A) One BCT (initial volume: 250 μ L) is composed of one million purified iPSC-derived non-treated CMs (>95% cTnT^{pos}) of either genotype (RAF1^{corr} and RAF1^{S257L}), 0.1 million γ -irradiated human foreskin fibroblasts and 10% Matrigel and 1 mg/mL rat tail collagen type I. **(B)** The culture medium of the treatment group of RAF1^{S257L}-BCTs was supplemented with 0.1 μ M PD0325901 from day 0. **(C)** To support tissue maturation, BCTs were stretched by 200 μ m increments every fourth day, starting on day 10 of tissue culture four times. **(D)** On day 27 or 28 of culture, tissue samples were subjected to terminal physiological analyses in multimodal bioreactor systems allowing for in-depth assessment of contractility, contractile kinetics, and calcium-sensitivity of BCTs **(E)**. Inset shows an individual culture vessel that is

equipped with two platinum electrodes for field stimulation and which is connected to a motor to apply precise preload and to an isometric force transducer to record contraction forces.

In the corresponding force measurements (new *Figure S7*), the tissue samples were paced by field stimulation (± 20 V, 5 ms) five times and, if necessary, at slightly higher frequencies than their spontaneous beating frequencies but never below 1 Hz. This was followed by an increase in preload of 200 μ m, with pacing repeated after an adaption phase. A total of six preload steps were applied, resulting in a total length increase of 1.2 mm. The values shown in the previous Figure 6B (now Figure 7B) are the forces at a preload step where the individual specimens reached their maximum contraction force (L_{max}). According to the Frank-Starling mechanism of preload-dependent increase in contraction force, maximum values between 0.8 mm - 1.2 mm preload were reached. Although some individual $RAF1^{S257L}$ BCTs started with higher spontaneous beat frequencies than 1 Hz at baseline, resulting in a mean frequency of 0.89 ± 0.2 Hz, the maximum contraction force was determined at preloads where the spontaneous beating frequency was below 1 Hz (maximum spontaneous beating frequency at L_{max} : 0.77 ± 0.18 Hz), i.e. at a field stimulation of 1 Hz. Therefore, all values for the maximum contraction force analysis were recorded under the same controlled pacing conditions (*Fig. S7*).

Figure S7. Preload-dependent behavior of BCT contractility. (A) Representative brightfield images of the three experimental groups before measurement on days 27 - 28 of culture. Scale bar: 1 mm. (B) Original traces of a representative measurement to determine the maximum contraction force of a BCT. Starting from a defined baseline length (BL) for all tissues, i.e. 6.8 mm, preload (red) is increased by 200 μ m increments and the spontaneous contractions (black) are recorded constantly. Before each preload step, the BCT is electrically paced five times at a frequency slightly above its spontaneous beating frequency but never below 1 Hz (yellow arrows). (C) Representative original traces of spontaneous contractions for each experimental group were recorded at baseline (left), 600 μ m (center), and 1.2 μ m (right) preload, respectively. (D) Spontaneous contraction frequencies of BCTs in response to increasing preload. (E) Preload-dependent physiological increase in paced contraction forces. For D and E: N = 18-26 individual tissue samples per group. Two-way ANOVA was applied. Error bars: \pm SEM. ***P<0.001 vs. baseline. Analysis of cardiomyocytes generated from the second individual carrying $RAF1^{S257L}$ (iPSC line 2).

We have completed the Materials and Methods section (page 3) and described and discussed it as a new supplementary Figure S7 (page 44).

Besides these overarching concerns, I have a few additional experimental concerns. Overall I find the work exciting but view it as requiring some more experimental evidence to be conclusive. Specific experimental critiques: My first two questions address the possibility that

phenotype changes seen here are simply due to line-to-line variability (ie, they could also be observed with another “wild type” line and are not due to RAF1). While the MEK inhibition studies partially address this possibility, additional experiments are needed:

Response: Thank you for pointing out this critical issue, which we have taken into account in our experimental designs. "Possible changes in phenotype due to line-to-line variability" is explained in detail in the response to your first comment. Briefly, we used three iPSC wild-type clones from three different genetic backgrounds (WT1-3), two RAF1^{S257L} clones (lines 1 and 2) from two unrelated individuals with Noonan syndrome in two different laboratories using the same protocols, and an isogenic corrected clone from RAF1^{S257L} line 2 (see Figure 1A), and analyzed extensively their molecular, cellular, and physiological properties with two types of 3D cardiac tissue models (CBs and BCTs). It is crucial to note that despite the genetic background diversity among the three wild-type clones employed, the results collectively display synergistically, distinct variances from the RAF1^{S257L} clones produced from two separate patients. The gene-corrected clone was specifically added to this study to confirm the observed disease-related phenotype differences between the wild-type versus RAF1 clones. The results confirmed the main findings of the study that monoallelic variation of the RAF1 gene is the cause of the observed disease-related differences in phenotype.

The majority of the reported phenotypes about MEK inhibition, however, could not fully reverse the disease phenotype due to the activation of parallel pathways and cross-talks along the onset of the disorder. Since many of the structural abnormalities and signaling imbalances that appear during different stages of cardiac development typically cannot be completely reversed, a full rescue of the disease development was not anticipated from the outset.

Another line of evidence for the minimal potential impact of clonal variability for the observed molecular and structural defects and rescue mechanism is the fact that samples and reprogrammed cell lines from two unrelated patients, each with a different genetic background, as well as primary cells from patient 1, consistently match the molecular signatures in line with the clinical manifestations of the patients.

Our coherent results clearly underline that there are no striking differences between the lines and the results are consistent between unrelated patient-derived iPSCs.

1) While I appreciate the vast amount of work that went into the physiology studies, at least some subset of these (calcium handling, tissue contractility) must be repeated in the isogenic control (RAF1_{iso}, currently described in Fig. 2F). This would go a long way toward providing conclusive evidence that the differences between WT and RAF1 mutant lines are truly reflective of RAF1 function and not line-to-line variability.

Response: Thank you for this valuable comment. As requested, we have extended the comparative physiological experiments of the RAF1^{S257L} line 2 with and without MEK inhibitor treatment by including samples generated from its gene-corrected isogenic control (RAF1^{corr}). For this purpose, we generated a large number of BCTs for physiological assessment. For reasons of clarity, we have excluded the WT data from these analyses as they are no longer relevant. As shown in *Figure R5* (a new Figure S7) the data confirmed our initial findings that the contractility, i.e., spontaneous contraction frequency and contractile tension, is significantly compromised by the studied RAF1 mutation. For the latter parameter, the reduction of almost 20% (4.37 ± 0.89 mN/mm² vs. 5.39 ± 0.24 mN/mm²) was completely rescued by MEK inhibition (5.43 ± 0.9 mN/mm²) (*Figure 7D*). Although this trend was also observed when compared to an unrelated WT group, our isogenic system shows highly significant evidence of hypertrophy in the untreated patient's myocardium, as indicated by significant differences in cross-sectional areas (0.81 ± 0.13 mm² vs. 0.99 ± 0.1 mm²) when rescued by MEK inhibition (0.8 ± 0.11 mm²) (*Figure 7A*). Furthermore, compared to RAF1^{corr} we observed a distinct effect of RAF1^{S257L} on contraction kinetics, which was partially rescued by the inhibition of MEK (*Figures 7E-H*).

As for calcium hypersensitivity, we could not detect this HCM-associated trait. However, we demonstrated a significant decrease in energy efficiency associated with $RAF1^{S257L}$, which is a hallmark of HCM (Figures 7I-K). Thus, we found that $RAF1^{S257L}$ -BCTs were unable to achieve maximal contractile force at low glucose concentrations, whereas gene-corrected control tissues did not exhibit this deficiency. This pathophysiological property was not positively affected by MEK inhibition that indicated MEK-independent control of the glucose-dependent-maximal contractile force.

Figure 7. Aberrant physiology of $RAF1^{S257L}$ BCTs and their response to inhibition of MEK. Measurements were conducted on days 27-28 of 3D tissue culture using isogenic $RAF1^{corr}$, $RAF1^{S257L}$, and $RAF1^{S257L} + MEKi$ BCTs. **(A)** Quantification of cross-sectional areas of BCTs derived from the different genotypes and treatments. **(B)** Spontaneous beating frequencies at baseline (BL) and at the preload step, where the maximum contraction force was recorded for each BCT, i.e. at L_{max} . **(C)** Maximum contraction forces of BCTs at L_{max} at 1 Hz electrical pacing. **(D)** Maximum contractile tensions based on the cross-sectional areas of BCTs. Contraction kinetics at L_{max} compared between **(E)** $RAF1^{corr}$ vs. $RAF1^{S257L}$ and **(F)** $RAF1^{S257L}$ vs. $RAF1^{S257L} + MEKi$. Quantification of **(G)** time to peak of the contraction and **(H)** time to 80% relaxation at L_{max} . Responses of BCT contractions to increasing calcium concentrations (range: 0.1 – 5 mM) in 5 mM glucose conditions normalized to contraction forces measured in high glucose (25 mM) medium compared between **(I)** $RAF1^{corr}$ vs. $RAF1^{S257L}$ and **(J)** $RAF1^{corr}$ vs. $RAF1^{S257L} + MEKi$. **(K)** Quantification of normalized

maximum contraction forces at 5 mM calcium under low glucose (5 mM) levels. **A-H**: N = 14-26 individual tissue samples per group. **I-K**: N = 5-8 individual samples per group. Depending on the presence of normally distributed values, one-way ANOVA or Kruskal-Wallis test was applied. Error bars for **A-D & G,H**: \pm SD; Error bars for **E,F & I,J**: \pm SEM. *P<0.05, #P<0.01, ***P<0.001, ****P<0.0001.

Accordingly, we have revised the former Results section "Negative impact of RAF1 S257L on the contractile apparatus" as follows (page 10):

The negative impact of RAF1^{S257L} on contraction physiology

Next, we analyzed the effect of RAF1^{S257L} on microtissue physiology using a multimodal bioreactor system to generate, cultivate, stimulate, and characterize BCTs. Compared to gene-corrected controls, RAF1^{S257L} BCTs showed a significant increase in the cross-sectional area and significantly increased beating frequency, with both parameters positively influenced by MEKi (Fig. 7A,B). Although no differences in absolute maximal contractile forces were observable in the mutated myocardium, contractile tension was significantly decreased (Fig. 7C,D). Again, treatment with the MEK inhibitor resulted in normalization to control values. Furthermore, significantly different contraction kinetics of RAF1^{S257L} myocardium, i.e. the shape of contraction, was adjusted by MEKi (Fig. 7E). The contraction velocity, indicated by the time to peak contraction (TTP), was significantly lower in the mutant myocardium, and MEKi effectively slowed this parameter down to control levels (Fig. 7G). The time to 80% relaxation was not significantly affected by either the mutation or MEKi, although a slight tendency to slow was observed in the latter group (Fig. 7H). We investigated the sensitivity of RAF1^{S257L} myocardium to calcium in a titration experiment in Tyrode's solution containing standard glucose concentration, i.e. 5 mM (Fig. 7I). Although no differences in responses to increasing calcium concentrations were observed between the groups, the contraction forces in the mutant myocardium did not reach the values measured in normal cultivation medium containing 25 mM glucose. This phenomenon, which points toward impaired energy efficiency, was not altered by MEKi (Fig. 7J,K).

Revision of the Discussion section "RAF1^{S257L} effects on cardiac excitation-contraction coupling" (pages 14):

RAF1^{S257L} effects on myocardial function

Expectedly, the physiology and contractile behavior of cardiac tissue was affected by the molecular alterations, structural abnormalities, and lower intracellular Ca²⁺ transients in RAF1^{S257L} CMs (Eisner *et al.*, 2017). In our isogenic model of miniaturized 3D myocardium, we observed different phenotypes and the effects of MEKi. In addition to the increase in tissue size, as determined by a 22 % larger cross-sectional area, our detailed analysis of myocardial physiology revealed that the spontaneous contraction frequency of RAF1^{S257L}-BCTs was increased by 157 % which is an indicator of the HCM state in the RAF1^{S257L}-BCTs and reduced contractile tension by 19 %. These three parameters were fully rescued by MEKi. In addition, the higher velocities of contraction kinetics in RAF1^{S257L}-BCTs, possibly caused by a stiffer PEVK region in titin, were also positively modulated by MEKi. Interestingly, this treatment did apparently not affect the potential loss in energy efficiency in the mutant myocardium. The phenomenon of impaired energy homeostasis is also known from other RASopathies, e.g., Costello syndrome, in which affected individuals also suffer from HCM (Kontaridis and Chennappan, 2022).

In addition, certain aspects of myocardial physiology were significantly altered by RAF1^{S257L}. Here, we have shown that deviant contractile properties such as spontaneous contraction frequency and contractile tension were positively influenced by the inhibition of MEK. However, the impaired energy inefficiency caused by RAF1^{S257L} was not affected by MEKi. Future studies will be needed to address this unaffected feature of RAF1^{S257L} myocardium. One promising approach may be to target the AMPK pathway, as has been described to modulate altered bioenergetics and reduce left ventricular hypertrophy in a mouse model of Costello syndrome (Dard *et al.*, 2022). In addition,

it will be important to determine whether it is possible to restore normal physiology after the RAF1^{S257L}-associated phenotype has manifested strongly. From a clinical perspective, systemic pharmacological intervention would usually be started postnatally when the hearts of affected individuals have already undergone developmental changes that significantly impair cardiac function.

2) While the studies here were only done with lactate-purified cardiomyocytes: before lactate, based on flow cytometry for cardiac troponin on dissociated cardiomyocytes, is the differentiation efficiency similar amongst the lines tested here? Are there cardiomyocyte markers that are very similar across all the iPSC-cardiomyocytes regardless of genotype? Or is there reason to believe there is a differentiation defect that may be unrelated to RAF1?

Response: Thank you very much for this reference. One of the most useful models for modeling congenital cardiac defects is patient-derived iPSCs. In several similar studies conducted, the iPSC-derived cardiac model also did not show cardiac differentiation efficiency before lactate selection or another purification method, since the scope of these studies does not focus on embryonic cardiac development but on the cardiac function that is impaired in affected individuals, peri- or postnatally. In our hands, after ten days of lactate selection, $11.6 \pm 3.7 \times 10^6$ cardiomyocytes per single starting iPSC were yielded on day 20 in RAF1^{S257L} differentiation, which is in line with efficiencies reported in the literature (Laco *et al.*, 2020). Hence, we did not experience a negative effect of RAF1^{S257L} on cardiomyogenic differentiation efficiencies. However, we cannot exclude that the efficiency of cardiac differentiation is different for each cell line. Therefore, we considered this critical point from the beginning and used at least two cell lines for each genotype (3 wild-types and 2 RAF1^{S257L} lines). In addition, we used the gene-corrected control for RAF1 and treated our RAF1^{S257L} line with MEKi. In this way, we emphasize that our observations are not due to differences between the cell lines.

Other experimental points:

3) A major point made early on is “A remarkable and unprecedented finding in the present study is the atypical I-bands, in both RAF1^{S257L} CMs as well as the heart biopsy samples from the corresponding individual with the heterozygous RAF1^{S257L} variant.” Data should be provided to back up this statement, or a reference to a previous study showing primary heart biopsy data.

Response: Thank you for this advice. When we observed this phenotype, we searched the literature for previous studies on human myocardium and found some similar ultrastructural studies on HCM samples (Maron and Ferrans, 1973; Maron *et al.*, 1974; Maron *et al.*, 1975). We inserted the following part to discuss similar studies that were performed in cardiac tissues of the HCM individuals (pages 13-14): “Consistent with our observed phenotype of the atypical I-band region in both the cardiac *in vitro* model and biopsy of the NS-RAF1 individual, Maron et al (1973) studied the ultrastructure of the myocardial biopsies from cardiomyopathies (Maron and Ferrans, 1973). They showed the EM pictures of the HCM patient that also lacks the I-band, whereas in other cases such as infundibular stenosis and aortic regurgitation, the I-band was visible. Another study investigated myocardium obtained from 22 patients with asymmetric septum hypertrophy. They also showed disarrayed sarcomere structures and reported some abnormalities in Z-bands. Although the authors did not mention directly that the I-band is not visible, they indicated some abnormalities in Z-bands. Notably, in their electron microscopy micrographs, we cannot see I-bands (Maron *et al.*, 1974). Additionally, Maron et al (1975) presented some EM pictures from hypertrophied cardiac tissue with the same phenotypes as their previous studies of the HCM specimens that altogether showed no I-band in their micrographs (Maron *et al.*, 1975). “

4) It is unfair to compare the contractility of tissues beating at different rates (Fig. 6B). I strongly recommend these either be repeated with electrical field pacing. If this is not possible, then at least the authors can report contractility only for the subset of tissues that have a very similar

spontaneous beating frequency (a significant number of tissues of each genotype and +/- MEK inhibitor appear to have a frequency near 0.75 Hz).

Response: We thank the reviewer for this constructive feedback. We hope that we have been able to address these important points in our response to your comment 2. Here, where we have described the conditions of these measurements in more detail, in particular, that all maximum contraction forces were determined at the same electrical pacing frequency, i.e. 1 Hz.

5) Differences in contractility kinetics (rise and relaxation times) should be assessed and reported – as in the comment above, this should only be done for tissues beating at the same or very similar rate, but ideally in paced tissues.

Response: Thank you for this comment. All the data regarding contractility kinetics have been recorded at the same electrical pacing frequency of 1 Hz. We have added a detailed method section for BCTs (page 5), which also includes the following: “The shape of contractions and contraction kinetics, i.e. time to contraction peak (TTP) and time to 80% relaxation (TTR) were assessed at the preload step, where the maximum contraction force was recorded for each sample, i.e. L_{max} with 1 Hz of electrical pacing.” Eschenhagen and Carrier recently reviewed that “Another parameter of the sarcomere function is the unloaded sliding velocity of thin filaments on immobilized S1-myosin. Several studies indicate that HCM mutations are associated with increased sliding velocity and dilated cardiomyopathy (DCM) with decreased sliding velocity” (Eschenhagen and Carrier, 2019).

Although the $RAF1^{S257L}$ mutation analyzed in our study is not a structural HCM mutation, we clearly showed its effects on the sarcomere organization as a potential overarching parameter. Hence, we have included the following text in the Discussion section (page 15): “Although the analyzed $RAF1^{S257L}$ mutation is not directly affecting sarcomeric proteins as most of the HCM-causing mutations, we observed its drastic impact on the sarcomere organization in CMs in patient myocardial biopsies and *in vitro*. Next to the potential impact of the dysfunctional titin composition as seen in $RAF1^{S257L}$ CMs, resulting in a loss of flexibility, the observed changes in contraction kinetics might be explained by an increased sliding velocity of thin filaments of the sarcomere, as caused by mutations in sarcomeric proteins in HCM (Sweeney *et al.*, 1998; Kawana *et al.*, 2017; Keller *et al.*, 2004). The aspect that in HCM the “sliding velocity” is increased might be useful to explain the contraction shapes and kinetics (Fig. 7 E-G).”

Minor experimental critiques:

1) For all physiology studies (calcium handling and contractility): please provide specific examples (e.g. representative calcium and force traces).

Response: Representative original traces of contraction forces for each experimental group are shown in the new Figure S7C.

2) Please show a representative BCT for each genotype in Figure 5, in part to clarify that these measurements reflect BCT sections and not cardiomyocytes plated into 2D.

Response: The iPSC-derived CBs (Figure 5G-I) were dissociated on day 40 of differentiation and grown on gelatin-coated coverslips for up to 7 days before loading with the fluorescent Ca^{2+} indicator Fura-2. On day 47, we performed the measurements for single cells (2D). Therefore, we did not apply BCTs here. We have inserted this information in the legend of the former Figure 5 (now Figure 6).

3) In panel 5G, two of the data points in the WT set seem to dominate the response. Have the authors performed an outlier analysis on these data?

Response: We double-checked the data. Neither ROUT nor Grubbs tests identified any outlier(s) in the data for peak height.

Writeup and discussion

1) Many of the features of the disease observed here, including lower calcium intake and hypocontractility, combined with larger cell size, are more similar to literature reports of dilated, rather than hypertrophic cardiomyopathy. This important difference should be highlighted and discussed in the context of other *in vitro* cardiomyopathy models (for example, dilated cardiomyopathy, and sarcomere-linked hypertrophic cardiomyopathy), especially those that have used iPSC.

Response: Thank you for this suggestion. We included other *in vitro* cardiomyopathy models in the Discussion section as follows (pages 15-16):

“Increased cell size is more indicative of HCM rather than an eccentric hypertrophic phenotype generally associated with DCM. We have observed an increase in the cell size of CBs (Figure 4E) and cross-sectional areas of BCTs (Figure 7A).

A very important point is that the majority of DCM cases are due to abnormalities of titin and sarcomere at the A-band region (Roberts *et al.*, 2015; Herman *et al.*, 2012). This is again a different phenotype from that observed in our disease model, with HCM abnormalities mainly affecting the I-band region of titin and the sarcomere which is a hallmark of HCM (Fig. 4). Hinson *et al.* confirm this critical difference using cardiac microtissues differentiated from human iPSCs with patient-specific A-band truncation variants of titin leading to pathogenic DCM phenotypes (Hinson *et al.*, 2015). On the other hand, the results in other studies using HCM-associated human iPSCs are consistent with the results of our study, not only in terms of the affected I-band region, but also in terms of other key HCM features such as sarcomere disorganization, cellular enlargements, and the central role of the hyperactivation of ERK1/2 signaling as a contributing factor to the development of HCM in models based on familial cardiomyopathy (Lan *et al.*, 2013; Wu *et al.*, 2019; Wang *et al.*, 2018; Davis *et al.*, 2016).

In addition, we found evidence for dysfunctional myocardial energy metabolism as another element contributing to the phenotype caused by RAF1^{S257L}. Similar abnormalities were recently described in an iPSC-based 3D model of sarcomere-linked HCM and were associated with myocardial dysfunction (Mosqueira *et al.*, 2018). These authors also observed higher spontaneous contraction frequencies, impaired contraction forces, and a switch from energy-hungry α MHC to energy-efficient β MHC. The latter has been reported for a variety of human iPSC-based HCM models, but not for DCM models (Eschenhagen and Carrier, 2019). Systolic dysfunction is more pronounced in DCM, whereas diastolic dysfunction is more pronounced in HCM (Wu *et al.*, 2019b). Consistent with this, we did not find impaired absolute contraction forces in BCTs, but an altered contractile tension (Figure 7C&D).

Another hallmark of HCM has elevated BNP levels (NPPB), which we also detected in RAF1^{S257L}-derived cardiac cells (Wu *et al.*, 2019b). Impaired calcium handling has also been reported in iPSC-derived CM of familial HCM (Lan *et al.*, 2013b).

There are opposing hypotheses on the effects of mutations on hyper- or hypocontractility (Ušaj *et al.*, 2022). However, hypocontractility and hypercontractility are described alternatively in the pathogenesis of HCM, depending on the gene- or mutation-specific effects, the cardiac model used, and the experimental design. In previous studies using an iPSC-derived model of HCM, both hyper-, and hypocontractility have been reported (Bhagwan *et al.*, 2020). Remarkably, most of the relevant data come from mutations of sarcomeric proteins, *e.g.*, MYH7, not in signaling components.

The pathogenesis we observed related to RAF1^{S257L}, such as disarrayed sarcomere structures, abnormalities in the I-bands, and shifts in titin isoforms, suggests that we may have changes in contractile behavior due to these defects (see Figure 7). However, we cannot say whether we had hypo- or hypercontractility by looking only at the maximum force development data in Figure 7C. Therefore, we prefer to leave this question open so that further studies with a more similar cardiac system can answer this question.”

2) Any disadvantages of mouse models of Noonan syndrome should be briefly provided as a rationale for using this system that produces human, albeit immature cardiomyocytes.

Response: Thank you for your constructive comment. We included the following text in the revised manuscript on page 16: "Despite the discovery of numerous mutations in genes underlying HCM and DCM, our understanding of the pathomechanisms leading from mutation to phenotype is still incomplete. Reasons for this are the diversity of mutations causing similar disease patterns, the incomplete and highly variable penetrance of HCM and DCM, and the fact that mouse models only partially recapitulate the human phenotype (reviewed in (Eschenhagen and Carrier, 2019). Despite the value of mouse models in the possibility to study the development of the effects of heart disease in the whole organism, there are some remarkable aspects of human models that cannot be studied in mice. For example, the resting heart rate of mice is ten times higher than that of humans. In mice, myosin heavy chain 6 (MYH6; faster isoform) is highly expressed in the ventricle, whereas in humans it is MYH7 (slower isoform). The development of the heart, the contribution of ion channels, and the electrical properties differ greatly between humans and mice (Vakrou et al., 2021). In any case, the experience with mouse models raises the question of the extent to which they truly reflect human disease and provides a strong argument to study HCM and DCM in human iPSC-derived CMs. The electrophysiology of the heart and cardiac cells, as well as their frequency and force of contraction, have a major impact on several significant physiological characteristics. Furthermore, RASopathy-associated HCM typically shows its greatest manifestation dynamics in the peri- or early postnatal period, suggesting a developmental phenotype. The maturity of our human iPSC-based 3D model seems to be sufficient to phenocopy many of the features that play a crucial role in the pathophysiology and progression of the disease and is a valuable tool to analyze the possibilities, but also the limitations, of MEK inhibition in terms of phenotype rescue *in vitro*."

References:

- Akerberg, B. N., Gu, F., VanDusen, N. J., Zhang, X., Dong, R., Li, K., Zhang, B., Zhou, B., Sethi, I., Ma, Q., Wasson, L., Wen, T., Liu, J., Dong, K., Conlon, F. L., Zhou, J., Yuan, G. C., Zhou, P. & Pu, W. T. (2019). A reference map of murine cardiac transcription factor chromatin occupancy identifies dynamic and conserved enhancers. *Nat Commun* 10(1): 4907.
- Dahlmann, J., Kensah, G., Kempf, H., Skvorc, D., Gawol, A., Elliott, D. A., Drager, G., Zweigerdt, R., Martin, U. & Gruh, I. (2013). The use of agarose microwells for scalable embryoid body formation and cardiac differentiation of human and murine pluripotent stem cells. *Biomaterials* 34(10): 2463-2471.
- Dard, L., Hubert, C., Esteves, P., Blanchard, W., Bou About, G., Baldasseroni, L., Dumon, E., Angelini, C., Delourme, M., Guyonnet-Dupérat, V., Claverol, S., Fontenille, L., Kissa, K., Séguéla, P. E., Thambo, J. B., Nicolas, L., Herault, Y., Bellance, N., Dias Amoedo, N., Magdinier, F., Sorg, T., Lacombe, D. & Rossignol, R. (2022). HRAS germline mutations impair LKB1/AMPK signaling and mitochondrial homeostasis in Costello syndrome models. *J Clin Invest* 132(8).
- Davis, J., Davis, L. C., Correll, R. N., Makarewich, C. A., Schwanekamp, J. A., Moussavi-Harami, F., Wang, D., York, A. J., Wu, H. & Houser, S. R. (2016). A tension-based model distinguishes hypertrophic versus dilated cardiomyopathy. *Cell* 165(5): 1147-1159.
- Eisner, D. A., Caldwell, J. L., Kistamas, K. & Trafford, A. W. (2017). Calcium and Excitation-Contraction Coupling in the Heart. *Circ Res* 121(2): 181-195.
- Eschenhagen, T. & Carrier, L. (2019). Cardiomyopathy phenotypes in human-induced pluripotent stem cell-derived cardiomyocytes-a systematic review. *Pflugers Arch* 471(5): 755-768.

- Gallo, S., Vitacolonna, A., Bonzano, A., Comoglio, P. & Crepaldi, T. (2019). ERK: A Key Player in the Pathophysiology of Cardiac Hypertrophy. *Int J Mol Sci* 20(9).
- Goldfracht, I., Efraim, Y., Shinnawi, R., Kovalev, E., Huber, I., Gepstein, A., Arbel, G., Shaheen, N., Tiburcy, M., Zimmermann, W. H., Machluf, M. & Gepstein, L. (2019). Engineered heart tissue models from hiPSC-derived cardiomyocytes and cardiac ECM for disease modeling and drug testing applications. *Acta Biomater* 92: 145-159.
- Herman, D. S., Lam, L., Taylor, M. R., Wang, L., Teekakirikul, P., Christodoulou, D., Conner, L., DePalma, S. R., McDonough, B. & Sparks, E. (2012). Truncations of titin causing dilated cardiomyopathy. *New England Journal of Medicine* 366(7): 619-628.
- Hinson, J. T., Chopra, A., Nafissi, N., Polacheck, W. J., Benson, C. C., Swist, S., Gorham, J., Yang, L., Schafer, S. & Sheng, C. C. (2015). Titin mutations in iPS cells define sarcomere insufficiency as a cause of dilated cardiomyopathy. *Science* 349(6251): 982-986.
- Kawana, M., Sarkar, S. S., Sutton, S., Ruppel, K. M. & Spudich, J. A. (2017). Biophysical properties of human β -cardiac myosin with converter mutations that cause hypertrophic cardiomyopathy. *Sci Adv* 3(2): e1601959.
- Keller, D. I., Coirault, C., Rau, T., Cheav, T., Weyand, M., Amann, K., Lecarpentier, Y., Richard, P., Eschenhagen, T. & Carrier, L. (2004). Human homozygous R403W mutant cardiac myosin presents disproportionate enhancement of mechanical and enzymatic properties. *J Mol Cell Cardiol* 36(3): 355-362.
- Kensah, G., Roa Lara, A., Dahlmann, J., Zweigerdt, R., Schwanke, K., Hegermann, J., Skvorc, D., Gawol, A., Azizian, A., Wagner, S., Maier, L. S., Krause, A., Drager, G., Ochs, M., Haverich, A., Gruh, I. & Martin, U. (2013). Murine and human pluripotent stem cell-derived cardiac bodies form contractile myocardial tissue in vitro. *Eur Heart J* 34(15): 1134-1146.
- Kontaridis, M. I. & Chennappan, S. (2022). Mitochondria and the future of RASopathies: the emergence of bioenergetics. *J Clin Invest* 132(8): 1-5.
- Laco, F., Lam, A. T.-L., Woo, T.-L., Tong, G., Ho, V., Soong, P.-L., Grishina, E., Lin, K.-H., Reuveny, S. & Oh, S. K.-W. (2020). Selection of human induced pluripotent stem cells lines optimization of cardiomyocytes differentiation in an integrated suspension microcarrier bioreactor. *Stem cell research & therapy* 11(1): 118.
- Lan, F., Lee, A. S., Liang, P., Sanchez-Freire, V., Nguyen, P. K., Wang, L., Han, L., Yen, M., Wang, Y. & Sun, N. (2013). Abnormal calcium handling properties underlie familial hypertrophic cardiomyopathy pathology in patient-specific induced pluripotent stem cells. *Cell stem cell* 12(1): 101-113.
- Maron, B. J. & Ferrans, V. J. (1973). Significance of multiple intercalated discs in hypertrophied human myocardium. *Am J Pathol* 73(1): 81-96.
- Maron, B. J., Ferrans, V. J., Henry, W. L., Clark, C. E., Redwood, D. R., Roberts, W. C., Morrow, A. G. & Epstein, S. E. (1974). Differences in distribution of myocardial abnormalities in patients with obstructive and nonobstructive asymmetric septal hypertrophy (ASH). Light and electron microscopic findings. *Circulation* 50(3): 436-446.
- Maron, B. J., Ferrans, V. J. & Roberts, W. C. (1975). Ultrastructural features of degenerated cardiac muscle cells in patients with cardiac hypertrophy. *Am J Pathol* 79(3): 387-434.
- Mosqueira, D., Mannhardt, I., Bhagwan, J. R., Lis-Slimak, K., Katili, P., Scott, E., Hassan, M., Prondzynski, M., Harmer, S. C. & Tinker, A. (2018). CRISPR/Cas9 editing in human pluripotent stem cell-cardiomyocytes highlights arrhythmias, hypocontractility, and energy depletion as potential therapeutic targets for hypertrophic cardiomyopathy. *European heart journal* 39(43): 3879-3892.
- Park, S. J., Choi, Y. S., Lee, S., Lee, Y. J., Hong, S., Han, S. & Kim, B. C. (2016). BIX02189 inhibits TGF- β 1-induced lung cancer cell metastasis by directly targeting TGF- β type I receptor. *Cancer Lett* 381(2): 314-322.

- Roberts, A. M., Ware, J. S., Herman, D. S., Schafer, S., Baksj, J., Bick, A. G., Buchan, R. J., Walsh, R., John, S. & Wilkinson, S. (2015). Integrated allelic, transcriptional, and phenomic dissection of the cardiac effects of titin truncations in health and disease. *Science translational medicine* 7(270): 270ra276-270ra276.
- Stein, C., Bardet, A. F., Roma, G., Bergling, S., Clay, I., Ruchti, A., Agarinis, C., Schmelzle, T., Bouwmeester, T., Schübeler, D. & Bauer, A. (2015). YAP1 Exerts Its Transcriptional Control via TEAD-Mediated Activation of Enhancers. *PLoS Genet* 11(8): e1005465.
- Sweeney, H. L., Feng, H. S., Yang, Z. & Watkins, H. (1998). Functional analyses of troponin T mutations that cause hypertrophic cardiomyopathy: insights into disease pathogenesis and troponin function. *Proc Natl Acad Sci U S A* 95(24): 14406-14410.
- Tatake, R. J., O'Neill, M. M., Kennedy, C. A., Wayne, A. L., Jakes, S., Wu, D., Kugler, S. Z., Jr., Kashem, M. A., Kaplita, P. & Snow, R. J. (2008). Identification of pharmacological inhibitors of the MEK5/ERK5 pathway. *Biochem Biophys Res Commun* 377(1): 120-125.
- Wang, L., Kim, K., Parikh, S., Cadar, A. G., Bersell, K. R., He, H., Pinto, J. R., Kryshtal, D. O. & Knollmann, B. C. (2018). Hypertrophic cardiomyopathy-linked mutation in troponin T causes myofibrillar disarray and pro-arrhythmic action potential changes in human iPSC cardiomyocytes. *Journal of molecular and cellular cardiology* 114: 320-327.
- Wu, H., Yang, H., Rhee, J.-W., Zhang, J. Z., Lam, C. K., Sallam, K., Chang, A. C., Ma, N., Lee, J. & Zhang, H. (2019). Modelling diastolic dysfunction in induced pluripotent stem cell-derived cardiomyocytes from hypertrophic cardiomyopathy patients. *European heart journal* 40(45): 3685-3695.

Responses to the comments of Reviewer #3

This manuscript presents a thorough and interesting investigation of the impact of a subtype of hypertrophic cardiomyopathies caused by mutations in the Raf protein (RASopathies). The authors developed several independent clones of human induced pluripotent stem cell lines from patients and healthy controls as well as an isogenic control using gene editing to repair the mutation and have used a 3D aggregate technique to differentiate these cells into cardiomyocytes in cardiac bodies. The authors demonstrated changes in MAPK signaling, downstream transcriptional changes of contractile proteins and calcium handling, and intriguing changes in the structure of the I-band, sarcomere, and z-disk. These structural changes in particular seem of interest to the field, and the addition of mechanics information from Bioartificial cardiac tissues and higher resolution structure from TEM imaging adds significant impact. They also provide significant mechanistic insights into the likely role of ERK signaling in regulating both transcriptional and downstream structural changes. There are a few clarifications mostly related to the impact of contractile molecular machinery which I believe could further strengthen this interesting manuscript.

Response: Thank you very much for reviewing our manuscript. My colleagues and I appreciate your positive and constructive comments, which are very valuable and important as they significantly increase the quality and clarity of our manuscript. We have responded to all comments point-by-point (see below). The most important changes are marked in cyan in the revised version of the manuscript.

Major comment

1. One of the most striking changes in transcription is the reduction in expression of alpha myosin heavy chain (MYH6) and a corresponding increase in beta myosin heavy chain (and associated light chains), which the authors discuss in the results and discussion. The authors mention the re-expression of fetal isoforms in reference to MYH7 on lines 445, 461, 468, and 581, but unlike in mice and small rodent models, MYH7 is the more dominant isoform of myosin in adult human ventricles, while MYH6 is the more prevalent fetal ventricular isoform.

Response: Thank you for pointing out these critical points and providing helpful references. We also agree that an increase in *MYH7* cannot be considered as “re-expression of fetal isoforms”, and we mainly mean the *NPPB*, as a fetal gene. To avoid misleading, we rephrased the sentences (page 11): “Accordingly, in *RAF1*^{S257L}-CBs compared to WT- and MEKi-treated CBs, we detected disruption in the expression pattern of contractile machinery proteins, including a transient shift from MYH6 to MYH7, in addition to calcium transient regulators.” and “The YAP-miR206-FOXP1 axis regulates CM hypertrophy and survival by modulating the expression patterns of important genes, e.g., *MYH7* and *NPPB*⁴⁶.”

On a related note, while alpha cardiac myosin (MYH6) does hydrolyze ATP at a rate 2-3 fold higher than beta cardiac myosin (MYH7) as mentioned on line 566, it is generally thought to generate less force due to its fast detachment under higher loads. The following reference provides some additional detail about the relative properties of these myosin isoforms. Cell Rep. 2015 May 12; 11(6): 910–920. Published online 2015 Apr 30. doi: 10.1016/j.celrep.2015.04.006 Ensemble Force Changes that Result from Human Cardiac Myosin Mutations and a Small-Molecule Effector; Tural Aksel,¹ Elizabeth Choe Yu,^{1,3} Shirley Sutton,¹ Kathleen M. Ruppel,^{1,2,*} and James A. Spudich^{1,*}

Response: Thank you for your expertise and your valuable comment. We add a new sentence on page 14: “The expression of MYH isoforms changes during normal heart development and disease progression. During human heart development, there is a shift from α -MHC to β -MHC. However, in end-stage HCM, there is generally even a higher transition in the ratio of β -MHC to α -MHC. An *in vitro* assay showed that α -MHC is faster and has a higher ATPase activity, and generates fewer forces than β -MYH (Aksel *et al.*, 2015).“

Some clarification of the identity of beta myosin as the dominant human adult form of myosin and the potential implications of what appears to be very low beta myosin expression in the control groups could clarify the relative maturity stage of these cardiomyocytes and the implications of these Raf mutations on maturation towards beta myosin expression.

Additionally, in the discussion of alpha and beta myosin isoforms expression, it would be appropriate to reference the prior work of Theresia Kraft, who has reported high alpha myosin expression in stem cell derived cardiac bodies relative to iPSC-CMs grown on stiffer substrates. [Stem Cell Reports Volume 14, Issue 5, 12 May 2020, Pages 788-80 Advanced Single-Cell Mapping Reveals that in hESC Cardiomyocytes Contraction Kinetics and Action Potential Are Independent of Myosin Isoform. The discussion of the alteration to contractile machinery could potentially be reframed to highlight the differences in sarcomere structure that might lead to impaired relaxation based on changes to titin and I-band leading into the discussion of calcium transient abnormalities and excitation-contraction coupling.

Response: it is generally accepted that iPSC-derived CMs are less mature compared to adult cardiac cells (Ahmed *et al.*, 2020). So, we expect more *MYH6* in iPSC-derived CMs in comparison to adult hearts. However, we could just check the ratio of the *MYH7/MYH6* in CBs and BCTs and not from samples of heart biopsy. Therefore, we cannot provide evidence that *MYH6* should be expressed at a higher ratio in iPSC-derived CMs compared to adult hearts. Also, the comparison of the different iPSC-derived CM models indicated that there is no unified consensus for various pathogenic conditions (Li *et al.*, 2022).

Concerning the relevancy of the observed decrease in *MYH7* expression in MEKi-treated CBs, we would like to point toward a recent comprehensive review of human iPSC-based *in vitro* cardiomyopathy models. Here, Eschenhagen and Carrier wrote “Three abnormalities appeared to be relatively consistent in both HCM and DCM—sarcomeric disarray ($274 \pm 81\%$, $n = 6$ HCM; $298 \pm 146\%$, $n = 8$ DCM), increased *NPPA* or *NPPB* gene expression ($284 \pm 249\%$, $n = 11$ HCM; 500% , $n = 2$) and arrhythmic behavior ($327 \pm 164\%$, $n = 12$ HCM; 350% , $n = 2$ DCM). HCM lines showed an **increase** in cell size ($156 \pm 85\%$, $n = 15$; DCM +/-), **in *MYH7* gene expression (or the ratio of *MYH7/MYH6* ($500 \pm 547\%$, $n = 8$; DCM +/- or reduction) and nuclear accumulation of the transcription factor NFAT ($175 \pm 65\%$, $n = 3$; DCM not determined).**” (Eschenhagen *et al.* Pflügers Archiv – European Journal of Physiology, 2019). Hence, the increase in *MYH7* expression that we observed, is consistent with data from the comparable models. Therefore, we added the following paragraph where we also cited the suggested work from Theresia Kraft (Weber *et al.*) as a reference for the comparatively low *MYH7* expression in iPSC-derived CBs (page 15): “In addition, we found evidence for dysfunctional myocardial energy metabolism as another element contributing to the phenotype caused by *RAF1*^{S257L}. Similar abnormalities were recently described in an iPSC-based 3D model of sarcomere-linked HCM and were associated with myocardial dysfunction (Mosqueira *et al.*, 2018). These authors also observed higher spontaneous contraction frequencies, impaired contraction forces, and a switch from energy-hungry α MHC to energy-efficient β MHC. The latter

has been reported for a variety of human iPSC-based HCM models, but not for DCM models (Eschenhagen and Carrier, 2019). Furthermore, a recent study reported a relatively low *MYH7* expression in human iPSC-derived CBs, which may explain our observation of pronounced changes in expression levels of this protein in *RAF1*^{S257L}-CBs in comparison to the wild-types and isogenic control lines (Weber et al., 2020). Systolic dysfunction is more pronounced in DCM, whereas diastolic dysfunction is more pronounced in HCM (Wu *et al.*, 2019). We did not find impaired absolute contraction forces in BCTs, but an altered contractile tension (Fig. 7C,D).”

2. The inclusion of BCT data highlights the value of more physiologic tension and linearized structure on the organization and maturation of these hiPSC-CMs. Given the discussion above, it would be very interesting to know the MYH7-MYH6 ratio of the BCT constructs by qPCR and/or immunostaining. Given the reported changes in titin isoform and length, it would also be interesting to determine if there were any differences in the passive stiffness between the control and mutant constructs. Finally, linearized 3D constructs have been shown to increase the maturity of calcium-handling machinery. It would be interesting to report if it is possible to observe calcium transients in the BCT constructs with either Fura 2 or a non-ratiometric reporter like Fluo4. Finally, given the presence of non-cardiomyocyte stromal cells in the BCT constructs, it is possible that the change in the cross-sectional area after MEKi treatment reflect a response of these stromal cells to this inhibition, which could be an additional comment in the discussion.

Response: Thank you for mentioning these very important points. Since BCT culture supports a higher level of CM maturity, we were interested in further investigating the MYH7-MYH6 ratio, titin isoform-associated changes in passive stiffness, and mutational changes in calcium handling in this 3D culture format. After analyzing the expression patterns of MYH7/MYH6 in the BCTs, we observed a distinct expression of MYH7 in the *RAF1*^{S257L} BCTs, which is consistent with our observations in CBs (*Figure R1*).

Figure R1. qPCR analysis of the MYH7-to-MYH6 ratio in BCTs. The data are averaged from three independent experiments in biological triplicates. *** $P < 0.001$, unpaired 2-tail t-test. $n=3$, biological replicates.

A comparison within isogenic conditions showed that the contraction kinetics of mutant BCTs are significantly faster. This could be a consequence of the presence of altered titin and indicative of a stiffened myocardium. Hence, we pointed out this observation in the Results section on page 10 by stating: “The contraction velocity, indicated by the time to peak of contraction (TTP), was significantly faster in the mutant myocardium, and MEKi effectively slowed this parameter down to control levels (Fig. 7G).” We also addressed it in the Discussion section on page 14 as follows: “In addition, the higher

velocities of contraction kinetics in RAF1^{S257L}-BCTs, possibly caused by a stiffer PEVK region in titin, were also positively modulated by MEKi.” Our future studies will certainly try to further address these important issues in more detail.

The mentioned non-cardiomyocyte stromal cells, i.e. HFF1, which were used to produce BCTs, were mitotically inactivated by gamma irradiation. Therefore, it is unlikely that they contribute to tissue thickness in mutant BCTs or respond to MEKi treatment in these tissues significantly. As this technical detail was not clear in the original submission, we have addressed this aspect in the Materials and Methods section as follows (page 5): “In brief, a mixture of rat tail collagen type I (Cultrex) and Matrigel™ (Life Technologies) was mixed with CMs and gamma-irradiated human foreskin fibroblasts (HFFs) and poured into silicone molds (5x5x10 mm, w/d/h) with two horizontal titanium rods that serve as suspensions for the developing tissue at a distance of 6 mm, resulting in BCTs with a volume of 250 µL each containing 10⁶ CMs, 10⁵ HFFs, collagen type I (230 µg), and 10% Matrigel.”

Minor comments

1. The timeframe of treatment of BCTs with MEKi is not clear.

Response: We agree that the start and duration of the MEKi treatment were not clear from the original manuscript and are grateful that the reviewer brought this to our attention. The MEKi treatment was started on day 0 of BCT culture and lasted until days 27-28 (i.e. days 48-49 of the start of iPSC cardiac differentiation) when the final measurements were performed.

We have taken this comment into account by changing the text passages in the Material and Methods section on page 5 as follows: “On day 21, non-MEKi-treated CBs were dissociated as described above and a previously described protocol was used to generate BCTs.” & “A culture medium composed of DMEM (25 mM glucose), 10 % horse serum, 2 mM L-glutamine (all Life Technologies), 10 µg/mL insulin, and 100 U/mL penicillin and 100 µg/mL streptomycin (Sigma-Aldrich) was exchanged every other day, and the medium for RAF1^{S257L} BCTs was additionally supplemented with 0.1 µM PD0325901.”

In addition, we have included a new supplementary Figure S6 in which we describe the generation, culture, treatment and measurement of BCTs in more detail.

2. In the methods the authors report using 1% volume dilution of B27 supplements which are produced at 50x concentration. If this is an accurate concentration, could the authors comment on the rationale for using a less-than-standard dilution factor.

Response: We thank the reviewer for pointing out this error. We used the diluted Supplement B-27 (with and without insulin) in our iPSC differentiation process. The section "Material and methods" in the manuscript was changed on page 3 as follows: “Differentiation was induced with the exchange of medium to RPMI 1640 supplemented with 1x B-27 supplement without insulin (RB⁻, Thermo Fisher Scientific, #A18956-01).” & “From d7-d10, EBs were cultivated with RPMI 1640 supplemented with 1x B-27 supplement with insulin (RB⁺, Thermo Fisher Scientific, #17504-044).”

References:

Ahmed, R. E., Anzai, T., Chanthra, N. & Uosaki, H. (2020). A Brief Review of Current Maturation Methods for Human Induced Pluripotent Stem Cells-Derived Cardiomyocytes. *Front Cell Dev Biol* 8: 178.

- Aksel, T., Choe Yu, E., Sutton, S., Ruppel, K. M. & Spudich, J. A. (2015). Ensemble force changes that result from human cardiac myosin mutations and a small-molecule effector. *Cell Rep* 11(6): 910-920.
- Eschenhagen, T. & Carrier, L. (2019). Cardiomyopathy phenotypes in human-induced pluripotent stem cell-derived cardiomyocytes-a systematic review. *Pflugers Arch* 471(5): 755-768.
- Li, J., Feng, X. & Wei, X. (2022). Modeling hypertrophic cardiomyopathy with human cardiomyocytes derived from induced pluripotent stem cells. *Stem Cell Res Ther* 13(1): 232.
- Mosqueira, D., Mannhardt, I., Bhagwan, J. R., Lis-Slimak, K., Katili, P., Scott, E., Hassan, M., Prondzynski, M., Harmer, S. C., Tinker, A., Smith, J. G. W., Carrier, L., Williams, P. M., Gaffney, D., Eschenhagen, T., Hansen, A. & Denning, C. (2018). CRISPR/Cas9 editing in human pluripotent stem cell-cardiomyocytes highlights arrhythmias, hypocontractility, and energy depletion as potential therapeutic targets for hypertrophic cardiomyopathy. *Eur Heart J* 39(43): 3879-3892.
- Weber, N. et al. Advanced Single-Cell Mapping Reveals that in hESC Cardiomyocytes Contraction Kinetics and Action Potential Are Independent of Myosin Isoform. *Stem Cell Rep* 14, 788–802 (2020).
- Wu, H., Yang, H., Rhee, J. W., Zhang, J. Z., Lam, C. K., Sallam, K., Chang, A. C. Y., Ma, N., Lee, J., Zhang, H., Blau, H. M., Bers, D. M. & Wu, J. C. (2019). Modelling diastolic dysfunction in induced pluripotent stem cell-derived cardiomyocytes from hypertrophic cardiomyopathy patients. *Eur Heart J* 40(45): 3685-3695.

REVIEWERS' COMMENTS:

Reviewer #1 (Remarks to the Author):

The authors have sufficiently addressed all my comments.

Reviewer #2 (Remarks to the Author):

The authors have done a nice job in responding to comments on the initial submission. I appreciate the improved clarity in the physiology studies and inclusion of significant new experimental data to validate genotype-phenotype relationship in the genome-edited line. I appreciate the authors' thoughtful answers to other critiques.

I do have some minor questions and clarifications that I hope the authors will address in their final writeup, from their methods section:

1) The authors mention "murine embryonic feeder cell-conditioned medium". Are they actually culturing the media with murine embryonic feeder cells? If so, for how long, what is the density of the feeder cells, etc. If they are not conditioning their media with culture on other cells, they should consider changing the name for better clarity.

2) The authors describe use of 1% B27 supplements in their differentiation and cardiomyocyte maintenance. This is non-standard in the field, where it is more typical to use 2% B27, and some investigators have even specified using 4%. If the authors are truly using 1% B27, they should mention that they use this non-standard formulation, and mention how they came to using this level. At this point in the manuscript revision process, either a supplemental figure, a citation of previous work, or short written description of how they validated the 1% B27 would be fine.

Reviewer #3 (Remarks to the Author):

This manuscript presents a thorough and interesting investigation of the impact of a subtype of hypertrophic cardiomyopathies caused by mutations in the Raf protein (RASopathies). The authors developed several independent clones of human induced pluripotent stem cell lines from patients and healthy controls as well as an isogenic control using gene editing to repair the mutation and have used a 3D aggregate technique to differentiate these cells into cardiomyocytes in cardiac bodies. The authors demonstrated changes in MAPK signaling, downstream transcriptional changes of contractile proteins and calcium handling, and intriguing changes in the structure of the I-band, sarcomere, and z-disk. These structural changes in particular seem of interest to the field, and the addition of mechanics information from Bioartificial cardiac tissues and higher resolution structure from TEM imaging adds significant impact. They also provide significant mechanistic insights into the likely role of ERK signaling in regulating both transcriptional and downstream structural changes.